# RankRAG: Unifying Context Ranking with Retrieval-Augmented Generation in LLMs

**Yue Yu** [*]
Georgia Tech

**Wei Ping** [*]
NVIDIA

**Zihan Liu**
NVIDIA

**Boxin Wang**
NVIDIA

**Jiaxuan You**
NVIDIA

**Chao Zhang**
Georgia Tech

**Mohammad Shoeybi**
NVIDIA

**Bryan Catanzaro**
NVIDIA

## Abstract

Large language models (LLMs) typically utilize the top-$k$ contexts from a retriever in retrieval-augmented generation (RAG). In this work, we propose a novel instruction fine-tuning framework RankRAG, which instruction-tunes a single LLM for the dual purpose of context ranking and answer generation in RAG. In particular, the instruction-tuned LLMs work surprisingly well by adding a small fraction of ranking data into the training blend, and outperform existing expert ranking models, including the same LLM exclusively fine-tuned on a large amount of ranking data. For generation, we compare our model with many strong baselines, including GPT-4-0613, GPT-4-turbo-2024-0409, and ChatQA-1.5, an open-sourced model with the state-of-the-art performance on RAG benchmarks. Specifically, our Llama3-RankRAG significantly outperforms Llama3-ChatQA-1.5 and GPT-4 models on nine knowledge-intensive benchmarks. In addition, it also performs comparably to GPT-4 on five RAG benchmarks in the biomedical domain without instruction fine-tuning on biomedical data, demonstrating its superb capability for generalization to new domains.

## 1 Introduction

Retrieval-augmented generation (RAG) (Lewis et al., 2020; Izacard & Grave, 2021; Lin et al., 2024; Wang et al., 2024) is a widely used technique for customizing large language models (LLMs) to handle long-tail knowledge (Mallen et al., 2023; Asai et al., 2024b), provide up-to-date information (Kasai et al., 2023), and adapt to specific domains and tasks (Xiong et al., 2024) without modifying the model weights. In general, a dense embedding-based retriever (Karpukhin et al., 2020; Lin et al., 2023; Wang et al., 2022) first retrieves top-$k$ chunked contexts from a collection of documents or external database for a given question. Then, LLM reads the top-$k$ contexts to generate the answer.

However, the current RAG pipeline has the following limitations: *i)* LLMs are not good at reading too many chunked contexts (e.g., top-100) even with the long-context window, not only due to efficiency reasons, but also because a shorter list of top-$k$ (e.g., 5, 10) contexts usually leads to higher accuracy of generation (e.g., see Table 5 in Xu et al., 2024b). *ii)* Given a small $k$, one needs a mechanism to ensure the *high recall* of relevant contents. Relying solely on a retrieval model may be inadequate due to challenges in learning effective local alignments across the entire embedding space to support accurate matching (Luan et al., 2021). In practice, a separate ranking model (Nogueira et al., 2020; Glass et al., 2022; Ma et al., 2023) that cross-encodes question and candidate context can work better than a dense embedding-based retriever for obtaining the most relevant top-$k$ contexts from top-$N$

---

[*]Yue Yu did this work during an internship at NVIDIA. Correspondence to: Yue Yu <yueyu@gatech.edu>, Wei Ping <wping@nvidia.com>.

candidates ($N \gg k$). *iii)* However, the zero-shot generalization capability of the expert ranking model can be relatively limited compared to the versatile LLM itself.

Based on the above considerations, our goal is to design an RAG instruction tuning pipeline that uses a single language model to achieve both high-recall context extraction and high-quality content generation. In previous study, instruction-tuned LLMs demonstrate a strong ability to extract answers from relevant context for a given question (e.g., OpenAI, 2023; Liu et al., 2024; Lin et al., 2024). This capability can be viewed as the "dual capability" of determining whether a chunk of context is relevant to the question thus is useful for generating the answer. We hypothesize that these capabilities mutually enhance each other. Motivated by this insight, we propose RankRAG, which instruction-tunes a single LLM for both context ranking and answer generation in the RAG framework. Furthermore, RankRAG expands upon existing instruction-tuning data by incorporating context-rich QA, retrieval-augmented QA and ranking datasets, enhancing the LLM's ability to filter out irrelevant contexts during both the retrieval and generation phases of RAG.

Our contribution can be summarized as follows:

- We propose RankRAG, a novel framework that enhances LLM's RAG capability through simultaneously instructing the LLM on context ranking and answer generation. During training, we design a specialized task focused on identifying relevant contexts or passages for a given question. This task is structured for ranking and framed as regular question answering with instruction, aligning more effectively with retrieval-augmented generation tasks. At inference, the LLM first reranks the retrieved contexts, then generates answer based on the refined top-$k$ (e.g., 5). This framework is readily applicable to diverse knowledge-intensive NLP tasks.

- Remarkably, we observe that integrating a small fraction of ranking data into the instruction tuning blend of LLM works surprisingly well on the evaluations of ranking associated with the RAG tasks, even surpassing the LLMs fine-tuned with $10\times$ more ranking data. We attribute this success to the transferable design of RankRAG training.

- We extensively compare the proposed RankRAG method with several strong baselines, including the open-sourced ChatQA-1.5. On nine general-domain and five biomedical knowledge-intensive benchmarks for RAG, Llama3-RankRAG-8B and Llama3-RankRAG-70B outperforms Llama3-ChatQA-1.5-8B and Llama3-ChatQA-1.5-70B by a margin, respectively.

In the remainder of the paper, we discuss related work in § 2. We introduce problem setup in § 3 and RankRAG method in § 4. We present the experimental setup in § 5, and conclude the paper in § 6.

## 2 Related Work

Retrieval-augumented generation (RAG) has been established for knowledge-intensive NLP tasks (Lewis et al., 2020; Borgeaud et al., 2022; Izacard et al., 2023; Izacard & Grave, 2021). In the standard process, a standalone dense-embedding-based retriever (e.g., Karpukhin et al., 2020) first retrieves relevant information from an external corpus, which the LLM then utilizes in the generation process. To improve this pipeline, recent research has focused on aligning retrievers to the needs of LLMs for generation (Shi et al., 2024; Lin et al., 2024), designing multi-step retrieval processes (Trivedi et al., 2023; Jiang et al., 2023; Jeong et al., 2024; Shao et al., 2023), or filtering irrelevant contexts (Wang et al., 2023c; Yoran et al., 2024; Xu et al., 2024a). To improve generation, several studies have designed instruction-tuning methods dedicated to enhancing the search (Ma et al., 2023; Zhu et al., 2024; Muennighoff et al., 2024) and RAG capability of LLMs (Liu et al., 2024; Lin et al., 2024; Luo et al., 2023; Asai et al., 2024a; Wang et al., 2024).

Although strong retrievers have been introduced (e.g., Lin et al., 2023; Yu et al., 2022; Wang et al., 2022, 2023a; Lee et al., 2024), one potential approach to improve retriever is optimizing it along with LLM in an end-to-end manner (e.g., Guu et al., 2020; Shi et al., 2024; Sachan et al., 2021; Izacard et al., 2023). However, this requires surrogate loss for optimization and complicates the training pipeline, especially when the embedding database needs to be re-indexed frequently due to the update of the embedding model (i.e., retriever).

Ranking serves as an intermediate step to improve the quality of information retrieval (Mitra et al., 2018), and has been applied to RAG pipeline for improving generation quality (Glass et al., 2022; Ram et al., 2023). However, these methods still rely on an additional moderate-sized model (e.g. BERT, T5) for ranking, which is often insufficient to capture the relevance between query and contexts

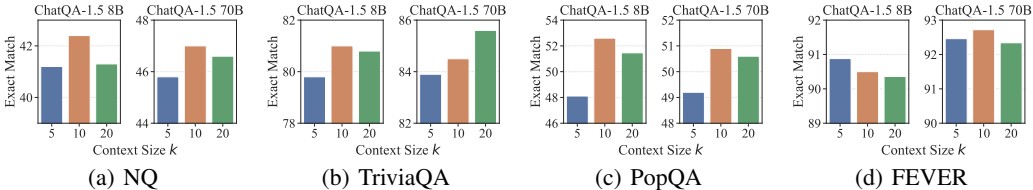

(a) NQ  (b) TriviaQA  (c) PopQA  (d) FEVER

Figure 1: Performance of ChatQA-1.5, one of the strongest RAG models, on different context size $k$. We observe a trade-off of selecting top-$k$ contexts: a smaller $k$ compromises the recall, while a larger $k$ could introduce irrelevant or noisy context and mislead the LLM generation.

and may lack the zero-shot generalization capability. Although recent studies have demonstrated the strong ability of LLMs at ranking tasks (Khalifa et al., 2023; Qin et al., 2024; Sun et al., 2023), how to harvest this ability for the RAG pipeline remains underexplored.

# 3 Preliminaries

In this section, we first introduce the preliminaries of retrieval-augmented generation as well as the problem setup. Then we present the limitations in the current RAG pipeline, which motivates the proposed RankRAG method.

## 3.1 Problem Setup

In retrieval-augmented generation, a collection of documents or contexts (e.g. Wikipedia) is given, providing the grounded knowledge. Given a question $q$, the retriever $\mathcal{R}$ (e.g., a parameterized embedding model) first retrieves top-$k$ contexts $\mathcal{C} = \{c_1, \cdots, c_k\}$ that are most relevant to the question. Subsequently, the language model produces the final answer where the answer can either be a short phrase or a long sentence, depending on the type of the target task. Our focus is on autoregressive language models (OpenAI, 2022, 2023; Meta-AI, 2024), which is the most common architectures for LLMs.

## 3.2 Limitation of Current RAG Pipelines

Before formally introducing RankRAG, we would like to first pinpoint several limitations of the current "retrieve-then-generate" pipeline with large language models.

**Limited Capacity of Retriever.** Current RAG systems usually employ sparse retrieval (e.g. BM25 (Robertson et al., 2004)) or moderate-size (e.g. BERT-based) embedding models (Karpukhin et al., 2020; Lin et al., 2023; Wang et al., 2022) as the retriever $\mathcal{R}$, mainly due to efficiency consideration as there are often millions of, if not more, documents need to be indexed. These models encode questions and documents *independently* and calculate the similarity between question and documents using vector similarity metrics. However, the *limited capacity of embedding models* and *independent processing of query and documents* constrain the ability to estimate textual relevance between question $q$ and documents $d$, reducing their effectiveness in new tasks or domains, verified by both theoretical (Menon et al., 2022) and empirical (Luan et al., 2021; Thakur et al., 2021) studies.

**Trade-off of Picking Top-$k$ Contexts.** Although the state-of-the-art long-context LLM can take many retrieved contexts as input for answer generation, the performance quickly saturates with increased $k$ in practice. For example, Xu et al. (2024b) finds the optimal number of chunked context $k$ is around 10 for long document QA tasks. As illustrated in Figure 1, we perform evaluation on ChatQA-1.5 (Liu et al., 2024), one of the strongest RAG models with open weights, and find the saturation of accuracy when $k = 10$. In general, a smaller $k$ often fails to capture all relevant information, compromising the *recall*, given the limited expressibility of retriever. In contrast, a larger $k$ improves *recall* but at the cost of introducing irrelevant content that hampers the LLM's ability to generate accurate answers (Yoran et al., 2024; Yu et al., 2023b).

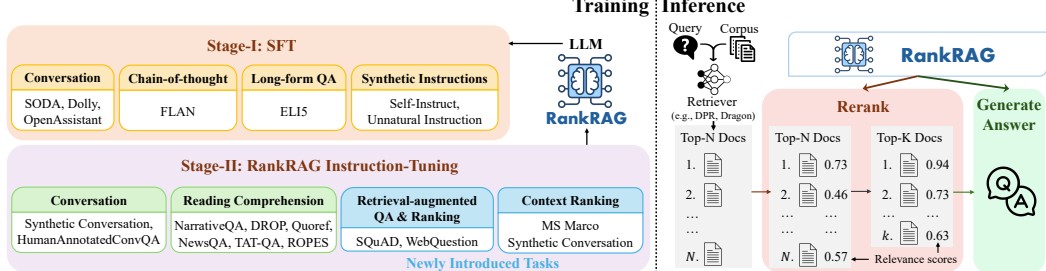

Figure 2: Two-stage instruction tuning framework for RankRAG.

# 4 RankRAG

To address the limitations mentioned in the previous section, we propose the RankRAG method to enhance the LLM's ability for retrieval-augmented generation. Specifically, we instruction-tune the LLM to simultaneously *capture the relevance between the question and context* and *utilize the retrieved context for answer generation*. The details are introduced as follows.

## 4.1 Stage-I: Supervised Fine-Tuning (SFT)

It is observed that general instruction-tuning or supervised fine-tuning (SFT) often significantly improves the ability of LLMs to follow instructions, thus improving zero-shot results on various downstream tasks (Wei et al., 2022; Ouyang et al., 2022). As such, we follow existing works (Chung et al., 2024; Wang et al., 2024; Liu et al., 2024) to first leverage SFT on a blend of high quality instruction following datasets, including: *i) a private crowd-sourced conversational dataset* and *public conversation datasets*: OpenAssistant (Köpf et al., 2023), Dolly (Conover et al., 2023), and SODA (Kim et al., 2023), *ii) a long-form QA dataset* ELI5 that requires elaborate answers (Fan et al., 2019), *iii) LLM-generated instructions*: Self-Instruct (Wang et al., 2023b) and Unnatural Instructions (Honovich et al., 2023), *iv) FLAN and Chain-of-thought datasets* (Chung et al., 2024).

There are overall 128K SFT examples in total. We make sure that there is *no overlap* between SFT data and data from evaluation tasks. For each sample in the instruction-following dataset, we take the multi-turn conversational format, use the previous turns of conversation between the user and the assistant as the context, and compute the loss only at the last response from the assistant.

## 4.2 Stage-II: Unified Instruction-Tuning for Ranking and Generation

The Stage-I SFT enpowers the LLMs with basic instruction-following capabilities; however, their performance on RAG tasks often remains suboptimal, as the LLMs are not optimized for extracting answers from retrieved context for a given question. Although recent studies (Lin et al., 2024; Liu et al., 2024; Zhang et al., 2024) enhance the RAG capability of LLM by instruction tuning it on context-rich generation tasks, these approaches can still be ineffective with poor initial retrieval results. RankRAG instruction tunes the LLM for both retrieval-augmented generation and context ranking. In particular, the context ranking capability is crucial to obtain more relevant top-*k* context with imperfect retriever.

To achieve this goal, the instruction tuning blend of Stage-II consists the following five parts:

1) **SFT data from Stage-I.** This part is included to maintain LLM's instruction-following capability.

2) **Context-rich QA data.** We first follow Liu et al. (2024) to leverage multiple context-rich QA tasks to enhance the LLM's capability of using context for generation. The training blend we use consists of: *i)* standard QA and reading comprehension datasets: DROP (Dua et al., 2019), NarrativeQA (Kočiský et al., 2018), Quoref (Dasigi et al., 2019), ROPES (Lin et al., 2019), NewsQA (Trischler et al., 2017), TAT-QA (Zhu et al., 2021), which contains a question, **a golden context** and an answer. *ii)* conversational QA datasets: HumanAnnotatedConvQA and SyntheticConvQA open-sourced by Liu et al. (2024), which contains a conversation between user and assistant, as well as one background document. The model needs to generate an answer given the conversation history and document.

3) **Retrieval-augmented QA data.** In addition to the above QA datasets used in Liu et al. (2024), we add two datasets with not only gold context but also the top-retrieved context using BM25. Note that it is crucial to improve LLM's robustness over irrelevant context at generation. Being aware of

Table 1: The instruction template for Stage-II. It is worth noting that all the tasks can be unified in the $(x, c, y)$ format, which is able to facilitate effective knowledge transfer across tasks.

| Task | Question $x$ | Context $c$ | Answer $y$ |
|---|---|---|---|
| Context-rich QA | Answer the following question from context. {question} | Passage: {Passage} (1 Psg.) | A phrase/sentence |
| Retrieval-augmented QA | Answer the following question from context. {question} | Passage 1: {Passage 1}... Passage 5: {Passage 5} (5 Psg. total) | A phrase/sentence |
| Context ranking | For the question {question}, access whether the passage is relevant to the question. | Passage: {Passage} (1 Psg.) | True/False |
| Retrieval-augmented ranking | For the question {question}, find all passages from the context that are relevant to the question. | Passage 1: {Passage 1}... Passage 5: {Passage 5} (5 Psg. total) | Passage Indexes |

this, we consider two QA tasks, namely SQuAD (Rajpurkar et al., 2016) and WebQuestions (Berant et al., 2013). For each question with the answer, we combine the gold context with the top-retrieved contexts using BM25, ensuring a total of five contexts. Note that some retrieved contexts may not contain the answer, and could be the "hard-negative" contexts.

4) **Context ranking data.** To empower LLMs with ranking capabilities, we use the popular MS MARCO passage (context) ranking dataset (Bajaj et al., 2016). We treat the gold query-passage pairs $(q, c^+)$ as relevant while using hard negative passages $(q, c^-)$ mined via BM25 as irrelevant pairs. The LLM needs to generate "True" or "False" given the corresponding query-passage pair, where the question along with the task-specific instruction is "For the question {question}, access whether the passage is relevant to the question.".

We want to handle ranking in conversational scenarios as well. While MS MARCO spans various topics, the questions are only single-turn short sentences. However, ranking data is only available, if any, at a small amount for conversation QA. To overcome this limitation, we repurpose the conversational QA pairs to generate pseudo relevance pairs. As each conversation is only associated with *one* document $d$, we cut each document into 150-word chunks $(c_1, c_2, \ldots, c_n)$. We compute the 4-gram recall score between each chunk $c_i$ and the ground-truth answer $a$, considering segments with a recall score above 0.5 as relevant and those below 0.1 as irrelevant for the corresponding conversation. Note that, each sample contains one question-context pair for this ranking dataset. In total, there are around 50k ranking pairs from MS MARCO ranking and synthetic conversations for instruction finetuning.

5) **Retrieval-augmented ranking data.** We aim to train the LLM with the capability of determining the relevance of multiple contexts simultaneously given a question, which is closer to the test-time behavior of RAG with top-$k$ contexts. As before, we use two QA datasets, SQuAD (Rajpurkar et al., 2016) and WebQuestions (Berant et al., 2013). We combine the gold context with the top-retrieved contexts using BM25, ensuring a total of five contexts. The contexts containing the answer are considered relevant, and the LLM is trained to *explicitly* identify all relevant contexts for the question.

**Unifying RAG and ranking with instruction tuning.** It is worth noting that, despite the variety of datasets and tasks described, they can all be cast into a standardized QA format $(x, c, y)$, where $x$ is the question, $c$ is the corresponding context, and $y$ is the target output answer. For example, for the retrieval-augmented ranking data, the question is "*For the question <question>, find all the passages from the context that are relevant to the question.*" Table 1 exhibits how to cast different tasks into a unified format. Despite its simplicity, this approach has the following advantages: *i*) It empowers the LLM with the ranking capability by adding a relatively small amount of ranking data. *ii*) By standardizing these tasks into a unified format, they can *mutually enhance* each other. After that, we obtain the final RankRAG model that can be applied to various knowledge-intensive NLP tasks.

### 4.3 RankRAG Inference: Retrieve-Rerank-Generate Pipeline

As RankRAG incorporates an additional reranking step, the inference pipeline for each question is modified as a *retrieve-rerank-generate* pipeline, described as follows: (1) the retriever $\mathcal{R}$ first retrieves top-$N$ contexts from the corpus. (2) the RankRAG model calculates the relevance score between the question and retrieved $N$ contexts as the probability of generating the answer as True using the prompt in Table 1, then reranks contexts to only retain top-$k$ ($k \ll N$) contexts, which are then used as the input for the generation step. (3) The top-$k$ contexts, along with the question, are concatenated and fed back into the RankRAG model to generate the final answer.

**Efficiency Discussion.** We are aware that the addition of a reranking step introduces extra processing time. In practice, for each question, denote the time for indexing and retrieval as $t_1$, the time for using LLM to calculate the relevance score as $t_2$ and the time for generation as $t_3$, then the ratio of added time overhead is $\frac{N*t_2}{t_1+t_3}$. In practice, calculating relevance typically requires generating just one token and involves much shorter inputs compared to the generation step with top-$k$ contexts. We provide efficiency study in §5.5.

## 5 Experiments

In this section, we conduct comprehensive experiments on a variety of knowledge-intensive NLP tasks to demonstrate the zero-shot capabilities of RankRAG.

### 5.1 Experiment Setup

**Tasks and Datasets.** We consider 3 types of tasks in experiments: (1) *Open-domain QA* (OpenQA), which includes NQ (Kwiatkowski et al., 2019), TriviaQA (Joshi et al., 2017), PopQA (Mallen et al., 2023), HotpotQA (Yang et al., 2018) and 2WikimQA (Ho et al., 2020). The first three are single-hop QA tasks, while the last two are multi-hop QA datasets. For NQ, TriviaQA, and HotpotQA, we use the split from KILT benchmark (Petroni et al., 2021) [2]. (2) *Fact verification*, where we use FEVER (Thorne et al., 2018) from KILT benchmark. (3) *Conversational QA* (ConvQA), we consider three datasets including Doc2Dial (Feng et al., 2020), TopiOCQA (Adlakha et al., 2022) and INSCIT (Wu et al., 2023), which have long documents that cannot be fitted directly into LLMs thus necessitates retrieval and ranking. The detailed dataset information is in Appendix A.1.

**Baselines.** We consider the following baselines: (1) *Baseline LLMs without RAG*, where we consider LLMs trained with proprietary data including InstructGPT (Ouyang et al., 2022), PaLM 2 (Anil et al., 2023), FLAN-LaMDA (Longpre et al., 2023), GLaM (Du et al., 2022), Claude 2 (Anthropic, 2023), Mixtral-8x22B-Instruct (Mistral, 2024), DeepSeek-V2 Chat (DeepSeek, 2024) and only use the official reported results. We also consider two ChatGPT-series models, namely GPT-3.5-turbo (gpt-3.5-turbo-0613) (OpenAI, 2022) and GPT-4 (gpt-4-0613) (OpenAI, 2023). (2) *Baselines with retrieval*, we evaluate models augmented with retrieval. Specifically, we include Atlas (Izacard et al., 2023) and Raven (Huang et al., 2023), two RAG models based on encoder-decoder LMs. For decoder-only models, we consider Self-RAG (Asai et al., 2024a), RECOMP (Xu et al., 2024a), InstructRetro (Wang et al., 2024), RePlug (Shi et al., 2024), RA-DIT (Lin et al., 2024), Llama-3-instruct (Meta-AI, 2024) and ChatQA-1.5 (Liu et al., 2024). We also list the result of RAG pipelines using InstructGPT (175B parameters) as the backbone including GenRead (Yu et al., 2023a), Retrieve-read (Lazaridou et al., 2022) and ReFeed (Yu et al., 2024), but mainly for reference. Other reported numbers are directly comparable if they follow the standard zero-shot settings.

**Evaluation Metrics.** For OpenQA datasets, we use *Exact Match (EM)* as the main metric but also report Accuracy for TriviaQA and PopQA and F1 score for HotpotQA and 2WikimQA as it is used in several studies (Asai et al., 2024a; Mallen et al., 2023). For FEVER, we use accuracy as the metric. For ConvQA datasets, we follow (Liu et al., 2024; Wang et al., 2024) to use F1 score as the metric.

**Implementation Details.** We use Llama3 8B and 70B (Meta-AI, 2024) as the backbone in our main experiments. For the two-stage instruction tuning, we set the batch size to 128 and train the model for 1000 steps with learning rate 5e-6 in Stage-I. Then, we reduce the learning rate to 3e-7 for 8B and 2e-7 for 70B model, set the batch size to 64, and train the model for 3300 steps (around 1 epoch). We use the Adam optimizer (Kingma & Ba, 2014) with $\beta_1 = 0.9$ and $\beta_2 = 0.98$. During the inference stage, we use the December 2018 Wikidump as the corpus index for NQ, TQA, HotpotQA, 2WikimQA, and use the December 2020 Wikidump for PopQA, following (Asai et al., 2024a). By default, we follow (Wang et al., 2024; Lin et al., 2024; Liu et al., 2024) to use the Dragon retriever (Lin et al., 2023) as default and retrieve top-$N$ (100 for 8B and 30 for 70B) documents for ranking, but RankRAG can be adapted to various retrievers and different $N$ (see § 5.3 and 5.5). To ensure a fair comparison, we test the performance of $k \in \{5, 10, 20\}$ and report *the best performance* for baselines. For generation, we keep temperature $T = 0$ and set the maximum number of generated token to be 32 for OpenQA, 128 for ConvQA and 8 for others. Training RankRAG-8B uses 32 NVIDIA A100 GPUs for 10 hours (4 hours for Stage-I and 6 hours for Stage-II finetuning), while training RankRAG-70B uses 128 NVIDIA A100 GPUs for 16 hours (4 hours for Stage-I and 12 hours for Stage-II Finetuning).

---

[2]The results of NQ and TriviaQA using the split from DPR (Karpukhin et al., 2020) are in Appendix F.

Table 2: Results of RankRAG and baselines on 9 datasets. Unless specified, all results are under ***zero-shot*** evaluation without additional demonstrations. Results unavailable in public reports are marked as "–". We use NQ, TriviaQA, and HotpotQA from the KILT benchmark for Llama3-Instruct, Llama3-ChatQA-1.5, and Llama3-RankRAG. Note that[†]: GPT-4 and GPT-4-turbo may refuse to answer the question when retrieved passages do not contain relevant information, thus the EM / accuracy drops after including RAG on TriviaQA, HotpotQA and 2WikimQA.

| Task (*Zero-shot*) | NQ | TriviaQA | PopQA | HotpotQA | 2WikimQA | FEVER | Doc2Dial | TopiOCQA | Inscit | Avg. |
|---|---|---|---|---|---|---|---|---|---|---|
| Metric | EM | EM / Acc. | EM / Acc. | EM / F1 | EM / F1 | Acc. | F1 | F1 | F1 | – |
| *Without Retrieval-Augmented Generation* | | | | | | | | | | |
| InstructGPT (Ouyang et al.) | 29.9 | 65.8 / 73.2 | – / – | 26.0 / 38.2 | 27.2 / 34.8 | 77.6 | – | – | – | – |
| PaLM2 540B (0 shot, Anil et al.) | 21.2 | 76.9 / – | – / – | – / – | – / – | – | – | – | – | – |
| PaLM2 540B (5 shot, Anil et al.) | 37.1 | 86.1 / – | – / – | – / – | – / – | – | – | – | – | – |
| GLaM 64B (0 shot, Du et al.) | 37.5 | 71.3 / – | – / – | – / – | – / – | – | – | – | – | – |
| FLAN-LaMDA 137B (Wei et al.) | 20.7 | 68.1 / – | – / – | – / – | – / – | – | – | – | – | – |
| Claude 2 (5 shot, Anthropic) | – | 87.5 / – | – / – | – / – | – / – | – | – | – | – | – |
| Mixtral-8x22B-Instruct (5 shot, Mistral) | 40.1 | 82.2 / – | – / – | – / – | – / – | – | – | – | – | – |
| DeepSeek-V2 236B (5 shot, DeepSeek) | 53.4 | 86.7 / – | – / – | – / – | – / – | – | – | – | – | – |
| GPT-3.5-turbo-1106 (OpenAI) | 38.6 | 82.9 / 91.7 | 28.4 / 32.2 | 29.9 / 42.0 | 23.9 / 30.4 | 82.7 | 20.1 | 28.5 | 27.2 | 38.5 |
| GPT-4-0613 (OpenAI) | 40.3 | 84.8 / 94.5 | 31.3 / 34.8 | 34.5 / 46.9 | 29.8 / 36.6 | 87.7 | 27.6 | 30.1 | 27.0 | 42.0 |
| GPT-4-turbo-2024-0409 (OpenAI) | 41.5 | 80.0 / 94.3 | 25.0 / 33.5 | 26.6 / 43.8 | 24.1 / 35.5 | 87.0 | 27.6 | 26.4 | 24.4 | 38.6 |
| *With Retrieval-Augmented Generation* | | | | | | | | | | |
| Atlas 11B (Izacard et al.) | 26.7 | 56.9 / – | – / – | 34.7 / – | – / – | 77.0 | – | – | – | – |
| Raven 11B (Huang et al.) | 29.6 | 65.7 / – | – / – | – / – | – / – | – | – | – | – | – |
| Self-RAG 7B (Asai et al.) | – | – / 66.4 | – / 54.9 | – / – | – / – | – | – | – | – | – |
| Self-RAG 13B (Asai et al.) | – | – / 69.3 | – / 55.8 | – / – | – / – | – | – | – | – | – |
| RECOMP 20B (Xu et al.) | 37.0 | 59.0 / – | – / – | 30.4 / 40.1 | – / – | – | – | – | – | – |
| InstructRetro 43B (Wang et al.) | 38.9 | 78.3 / – | – | – / – | – / – | – | 36.0 | – | – | – |
| RePlug 65B (Shi et al.) | 28.8 | 72.6 / – | – / – | 32.0 / – | – / – | 73.3 | – | – | – | – |
| RA-DIT 65B (Lin et al.) | 35.2 | 75.4 / – | – / – | 39.7 / – | – / – | 80.7 | – | – | – | – |
| Llama3-Instruct 8B (Meta-AI) | 30.9 | 70.7 / 80.4 | 34.9 / 55.8 | 26.0 / 35.8 | 9.6 / 25.2 | 88.9 | 33.6 | 44.9 | 32.6 | 40.8 |
| Llama3-Instruct 70B (Meta-AI) | 42.7 | 82.4 / 89.3 | 45.3 / 56.4 | 35.5 / 43.3 | 13.5 / 27.9 | 91.4 | 37.9 | 49.7 | 36.2 | 47.1 |
| Llama3-ChatQA-1.5 8B (Liu et al.) | 42.4 | 81.0 / 87.6 | 52.6 / 59.8 | 33.4 / 44.6 | 26.8 / 31.9 | 90.9 | 39.3 | 49.9 | 30.1 | 49.6 |
| Llama3-ChatQA-1.5 70B (Liu et al.) | 47.0 | 85.6 / 91.4 | 50.9 / 58.3 | 42.2 / 54.4 | 34.9 / 37.4 | 92.7 | 41.3 | 55.6 | 32.3 | 53.6 |
| Llama3-RankRAG 8B (0 shot) | 50.6 | 82.9 / 89.5 | 57.6 / 64.1 | 35.3 / 46.7 | 31.4 / 36.9 | 92.0 | 40.4 | 50.4 | 33.3 | 52.6 |
| Llama3-RankRAG 70B (0 shot) | 54.2 | 86.5 / 92.3 | 59.9 / 65.4 | 42.7 / 55.4 | 38.2 / 43.9 | 93.8 | 41.5 | 52.8 | 35.2 | 56.1 |
| **For reference**: *Using InstructGPT or CodeX (~175B) (Ouyang et al., 2022) as the Backbone LLM.* | | | | | | | | | | |
| GenRead (Yu et al.) | 32.5 | 66.2 / – | 46.0 / – | 36.4 / 39.9 | – / – | 80.4 | – | – | – | – |
| Retrieve-Read (Lazaridou et al.) | 31.7 | 61.4 / – | – / – | 35.2 / 38.0 | 27.7 / – | 82.7 | – | – | – | – |
| ReFeed (Yu et al.) | 39.6 | 68.9 / – | – / – | 41.5 / 45.1 | – / – | – | – | – | – | – |
| GPT-3.5-turbo-1106 RAG (OpenAI) | 46.7 | 79.7 / 88.0 | 49.9 / 57.0 | 31.2 / 41.2 | 27.2 / 32.2 | 90.8 | 34.8 | 44.3 | 35.3 | 46.8 |
| GPT-4-0613 RAG[†] (OpenAI) | 40.4 | 75.0 / 88.5 | 44.3 / 61.4 | 27.6 / 38.1 | 14.4 / 17.6 | 92.6 | 34.2 | 45.1 | 36.4 | 43.5 |
| GPT-4-turbo-2024-0409 RAG[†] (OpenAI) | 40.3 | 70.2 / 91.1 | 39.5 / 58.4 | 8.1 / 17.9 | 22.8 / 39.2 | 92.2 | 35.4 | 48.3 | 33.8 | 41.6 |

**Data Contamination Issues.** One possible issue for the zero-shot evaluation is the test set contamination, where some of the task-specific examples overlap with the instruction fine-tuning data (Oren et al., 2024). To address this issue, we have performed a string match-based analysis where we do not observe any overlap between the train data and data from target tasks.

## 5.2 Main Experiments

Table 2 presents results of RankRAG and baselines. The findings are summarized as follows:

**RankRAG outperforms existing RAG methods.** With 8B scale, RankRAG consistently outperforms ChatQA-1.5 8B, one of the most recent open-sourced models with state-of-the-art performance on many RAG benchmarks. RankRAG 8B is also competitive when compared with baseline models with many more parameters. For example, it significantly outperforms InstructRetro (5× parameters), RA-DIT 65B (8× parameters), and even outperforms Llama3-instruct 70B (8× parameters) on NQ and TriviaQA tasks. With more parameters, RankRAG 70B outperforms the strong ChatQA-1.5 70B model, and largely outperforms previous RAG baselines with InstructGPT as the underlying LLM.

**RankRAG demonstrates larger improvement on more challenging datasets.** We observe that the performance gains of RankRAG over baselines are more pronounced for more challenging QA datasets. For example, on long-tailed QA (PopQA) and multi-hop QA (2WikimQA) tasks, we achieve more than 10% improvement over ChatQA-1.5. These findings suggest that in challenging OpenQA datasets where top documents from retrievers are less relevant to the answer, context ranking effectively enhances performance. In this work we focus on improving single-time retrieval for QA tasks. How to effectively combine multi-round RAG pipelines (Jiang et al., 2023; Khattab et al., 2022; Jeong et al., 2024) with RankRAG is an interesting avenue of future work.

## 5.3 Ablation Studies

**Effect of Designed Components.** Table 3 shows the ablations of RankRAG with Llama3 8B as the backbone on nine general-domain datasets. Overall, we observe all of the proposed components

Table 3: Ablation study of RankRAG. We use Llama3-8B as the backbone. Where 'RQA' and 'RAR' stands for retrieval-augmented QA and retrieval-augmented ranking data, respectively. For 'w/o reranking', we do not perform ranking in the inference stage.

| Task (Zero-Shot) | NQ | TriviaQA | PopQA | HotpotQA | 2WikimQA | FEVER | Doc2Dial | TopiOCQA | Inscit | Avg. |
|---|---|---|---|---|---|---|---|---|---|---|
| Metric | EM | EM / Acc. | EM / Acc. | EM / F1 | EM / F1 | Acc. | F1 | F1 | F1 | – |
| RankRAG 8B | **50.6** | **82.9 / 89.5** | **57.6 / 64.1** | 35.3 / **46.7** | 31.4 / 36.9 | 92.0 | **40.4** | **50.4** | 33.3 | **52.6** |
| w/o reranking | 48.0 | 80.3 / 86.8 | 49.3 / 59.0 | 31.3 / 41.6 | 26.4 / 30.5 | 91.1 | 39.7 | 49.4 | 30.9 | 49.8 |
| w/o RQA | 49.4 | 82.0 / 88.9 | 55.1 / 62.9 | **35.6** / 45.9 | **31.8 / 37.5** | **92.1** | 39.4 | 46.8 | 32.4 | 51.6 |
| w/o RAR | 48.6 | 82.2 / 89.1 | 56.0 / 62.6 | 35.1 / 45.2 | 31.2 / 35.7 | 91.4 | 39.6 | 48.6 | **33.5** | 51.8 |
| w/ RAFT (Lin et al.) | 43.3 | 80.8 / 87.6 | 48.9 / 56.3 | 30.5 / 41.8 | 25.2 / 29.6 | 91.2 | 36.6 | 46.4 | 30.1 | 48.1 |
| w/ Stage-I SFT Only | 38.3 | 63.7 / 76.6 | 49.8 / 54.6 | 26.5 / 40.3 | 18.0 / 25.9 | 85.7 | 33.3 | 33.7 | 30.5 | 42.2 |

Table 4: Zero-shot evaluation using Llama2 (Touvron et al., 2023) model as the backbone.

| Task (Zero-Shot) | NQ | TriviaQA | PopQA | HotpotQA | 2WikimQA | FEVER | Doc2Dial | TopiOCQA | Inscit | Avg. |
|---|---|---|---|---|---|---|---|---|---|---|
| Metric | EM | EM / Acc. | EM / Acc. | EM / F1 | EM / F1 | Acc. | F1 | F1 | F1 | – |
| Llama2-70B (Touvron et al.) | 25.3 | 82.4 / – | – / – | – / – | – / – | – | – | – | – | – |
| Llama2-ChatQA-1.0 7B (Liu et al.) | 41.4 | 77.8 / 86.5 | 46.7 / 55.0 | 28.9 / 40.3 | 24.0 / 27.5 | 85.9 | 37.9 | 45.5 | 31.0 | 46.6 |
| Llama2-ChatQA-1.0 13B (Liu et al.) | 47.9 | 80.9 / 87.6 | 51.8 / 56.2 | 32.9 / 43.2 | 27.6 / 31.1 | 87.6 | 38.1 | 48.9 | 30.8 | 49.6 |
| Llama2-ChatQA-1.0 70B (Liu et al.) | 49.5 | 83.2 / 89.7 | 52.1 / 56.6 | 39.0 / 49.4 | 28.9 / 34.1 | 91.7 | 38.9 | 51.0 | 31.9 | 51.8 |
| Llama2-RankRAG 7B | 46.9 | 84.0 / 89.6 | 55.9 / 61.3 | 32.2 / 43.2 | 26.8 / 30.7 | 86.6 | 38.6 | 49.2 | 32.3 | 50.3 |
| Llama2-RankRAG 13B | 50.5 | 84.5 / 91.0 | 58.0 / 63.9 | 36.4 / 47.3 | 29.5 / 34.2 | 91.7 | 39.5 | 49.2 | 33.4 | 52.5 |
| Llama2-RankRAG 70B | 53.2 | 85.8 / 92.1 | 58.7 / 64.5 | 41.8 / 53.1 | 33.8 / 38.8 | 91.9 | 41.2 | 52.9 | 35.8 | 55.0 |

contribute to the final performance. Removing context ranking hurts performance on all tasks, justifying its efficacy in selecting the most relevant contexts for the target question. Besides, the retrieval-augmented QA (RQA) and retrieval-augmented ranking (RAR) designed for instruction fine-tuning improve outcomes on most tasks by helping the model explicitly pinpoint relevant contexts. On the contrary, the RAFT method used in (Lin et al., 2024) treats each retrieved context separately during instruction finetuning, which yields suboptimal results when compared to RankRAG with the same training data.

**Performance with Different LLMs.** Table 4 reports the performance of RankRAG and the most recent baseline ChatQA using Llama2 with backbone having varying amounts of parameters. Notably, there exist consistent gains in terms of the average performance (7.8%/6.4%/6.3% on 7B/13B/70B variants respectively), justifying the advantage of RankRAG across different LLM types and scales.

**Performance with Different Retrievers.** Figure 3 exhibits the performance of RankRAG and ChatQA-1.5 with different dense retrievers on three representative tasks, where we consider DPR (Karpukhin et al., 2020) and Contriever-MS MARCO (Izacard et al., 2022) as two variants. We note that although the initial retrieved result is not good enough, RankRAG still surpasses ChatQA-1.5 by more than 10% for both retrievers on average. To summarize, RankRAG is robust to the choice of retrievers.

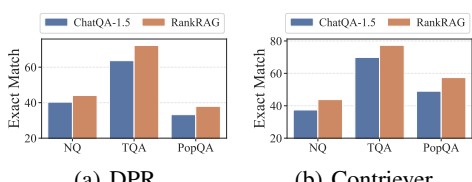

(a) DPR        (b) Contriever

Figure 3: Performance with different retrievers. The performance of Recall is in Appendix E.1.

### 5.4 Experiment on Domain-specific RAG Benchmarks

To demonstrate that RankRAG can adapt to specialized domains, we conduct experiments on Mirage (Xiong et al., 2024), a recently introduced RAG benchmark for the biomedical field. We follow Xiong et al. (2024) to employ MedCPT (Jin et al., 2023) as the retriever $\mathcal{R}$ with MedCorp[3] as the corpus $\mathcal{D}$.

Table 5: The performance of RankRAG on Mirage, a zero-shot biomedical RAG benchmark. RankRAG and baselines use retrieval by default. Most of numbers are from (Xiong et al., 2024).

| Datasets | MMLU-med | PubmedQA | BioASQ | MedQA | MedMCQA | Avg. |
|---|---|---|---|---|---|---|
| GPT-4-0613 (OpenAI) | 87.24 | 70.60 | 92.56 | 82.80 | 66.65 | 79.97 |
| GPT-3.5 (OpenAI) | 75.48 | 67.40 | 90.29 | 66.61 | 58.04 | 71.56 |
| Mixtral 8*7B (Jiang et al.) | 75.85 | 67.60 | 87.54 | 60.02 | 56.42 | 69.49 |
| Llama2 70B (Touvron et al.) | 54.55 | 50.40 | 73.95 | 44.93 | 43.08 | 53.38 |
| Meditron 70B (Chen et al.) | 65.38 | 56.40 | 76.86 | 49.57 | 52.67 | 60.18 |
| PMC-llama 13B (Wu et al.) | 52.53 | 42.58 | 48.29 | 56.00 | 65.21 | 52.92 |
| Llama3-ChatQA-1.5 8B (Liu et al.) | 61.40 | 66.40 | 82.69 | 42.36 | 46.97 | 59.96 |
| Llama3-ChatQA-1.5 70B | 80.51 | 74.80 | 83.17 | 68.89 | 62.54 | 73.98 |
| Llama3-RankRAG 8B | 64.55 | 65.00 | 84.44 | 48.86 | 56.90 | 63.95 |
| Llama3-RankRAG 70B | 81.44 | 79.80 | 90.76 | 69.21 | 69.11 | 78.06 |

The experiment results of RankRAG and baselines are shown in Table 5. From the table, we observe that RankRAG, even without fine-tuning on the biomedical domain, excels at medical QA tasks.

---

[3]Link: `https://huggingface.co/MedRAG`. Detailed dataset information is in Appendix A.2.

Table 6: Ranking performance with different ranking models. Unless specified, all baselines are used to rank the top 100 retrieved passages. RankRAG achieves better performance despite using fewer ranking data. * NQ, TriviaQA and HotpotQA are used for training the BGE-Reranker model. [†]: Our re-implementation. [‡] We only rerank top-30 passages for GPT-4 due to budget constraint.

| Task | # Rank Data | NQ | | | TriviaQA | | | PopQA | | | HotpotQA | | | Inscit | | |
|---|---|---|---|---|---|---|---|---|---|---|---|---|---|---|---|---|
| Recall | | R@5 | R@10 | R@20 | R@5 | R@10 | R@20 | R@5 | R@10 | R@20 | R@5 | R@10 | R@20 | R@5 | R@10 | R@20 |
| **Backbone Retriever** | | | | | | | | | | | | | | | | |
| Dragon (Lin et al.) | – | 74.9 | 80.3 | 84.3 | 89.0 | 92.9 | 95.3 | 69.6 | 76.9 | 82.6 | 47.5 | 52.4 | 60.1 | 43.4 | 56.0 | 64.9 |
| **Finetuned Baseline Ranking Model** | | | | | | | | | | | | | | | | |
| RankBERT 110M (Glass et al.) | ∼503k | 73.5 | 79.3 | 84.0 | 88.4 | 92.0 | 95.5 | 78.7 | 82.8 | 85.5 | 54.6 | 59.8 | 63.7 | 45.6 | 57.1 | 66.7 |
| monoT5 3B (Nogueira et al.) | ∼503k | 75.6 | 80.9 | 84.9 | 90.7 | 93.6 | 95.9 | 81.0 | 83.6 | 85.9 | 54.8 | 60.2 | 63.3 | 48.6 | 59.4 | 68.8 |
| BGE-Rerank-v2-m3 568M (Chen et al.) | ∼1.6M | 78.0* | 82.8* | 85.6* | 91.6* | 94.5* | 97.1* | 79.6 | 84.5 | 86.9 | 58.5* | 61.8* | 65.0* | 51.3 | 59.8 | 69.7 |
| RankLLaMA 8B[†] (Ma et al.) | ∼503k | 77.8 | 83.1 | 86.0 | 91.2 | 93.1 | 96.4 | 80.1 | 84.3 | 86.8 | 57.1 | **62.1** | 64.8 | 57.8 | 62.1 | 71.3 |
| ChatQA-1.5 8B (Liu et al.) | N/A | 68.2 | 75.7 | 82.0 | 85.4 | 91.1 | 94.0 | 67.3 | 76.7 | 83.5 | 37.4 | 45.0 | 53.6 | 32.3 | 42.6 | 54.9 |
| **Off-the-shelf LLM Reranker** | | | | | | | | | | | | | | | | |
| GPT-3.5 (**top 100**, OpenAI) | Unk. | 77.8 | 82.5 | 85.7 | 91.1 | 94.4 | 96.7 | 77.4 | 82.0 | 85.5 | 52.1 | 56.6 | 62.4 | 50.2 | 59.1 | 68.6 |
| GPT-4[‡] (**top 30**, OpenAI) | Unk. | 79.3 | 83.2 | 85.1 | 92.8 | 95.5 | 96.8 | 79.3 | 83.6 | 86.2 | 53.2 | 57.0 | 61.0 | 52.3 | 61.7 | 70.0 |
| **Our Model** | | | | | | | | | | | | | | | | |
| RankRAG 8B (**top 100**) | ∼50k | 80.3 | **84.0** | **86.3** | 93.2 | 95.4 | **97.3** | 81.6 | **84.9** | **87.0** | **57.6** | 61.8 | **65.2** | 60.9 | 65.7 | 73.5 |
| RankRAG 70B (**top 30**) | ∼50k | **80.6** | **84.0** | 85.4 | **93.6** | **95.9** | 97.1 | **81.8** | 84.6 | 86.5 | 56.3 | 59.7 | 62.2 | **61.3** | **66.4** | **74.6** |

Notably, RankRAG 8B surpasses Meditron 70B—a leading open-source LLM for the medical domain—by 6.3%. Besides, RankRAG 70B attains more than 98% performance of GPT-4. These results justify RankRAG's capacity to be readily applied to new domains without extra post-training.

## 5.5 A Closer Look at the Ranking Module

As the context ranking serves as a core step in RankRAG, we take a closer look at this component. All the studies are done using Llama3-8B as the backbone.

**RankRAG is Data-efficient.** Previous approaches that infuse context ranking into the RAG pipeline usually involve a separate reranking model. To compare our model with these baselines, we evaluate four models (BERT (Glass et al., 2022)/T5 (Nogueira et al., 2020)/Llama3 (Ma et al., 2023)) fine-tuned on the full MS MARCO passage ranking dataset, a strong off-the-shelf reranker model BGE-ranker, and two OpenAI GPT-series models. For the GPT-series models, we use the token probability of 'True' as a proxy for the relevance score[4]. These models are then used to rerank top-retrieved passages by Dragon, similar to our approach. Surprisingly, as shown in Table 6, RankRAG achieves better recall over dedicated ranking models trained on $10\times$ more ranking data for most cases. Besides, RankRAG can still outperform the BGE-ranker on most tasks, which has been extensively trained on more than 1 million ranking pairs, including some that overlap with our evaluation tasks. This advantage is likely due to the adaptable nature of our model's training, where the ranking data closely resembles the general RAG fine-tuning data. Directly using ChatQA-1.5 to rank passages *hurts* the performance, indicating the necessity of incorporating ranking data into instruction fine-tuning.

We further study the relation between the number of context ranking data and final performance. As shown in Figure 4, with 5k ranking data only ($\sim 1\%$ of the MS MARCO dataset), RankRAG can already obtain very compelling results, while further increasing the number of ranking data to 50k yields non-marginal gains. This finding confirms RankRAG's data efficiency – achieving effective performance with a modest amount of ranking data and maintaining adaptability across various tasks.

**Performance v.s. Time-efficiency for RankRAG.** One specific caveat for scaling up model size is the increment in the latency overhead — as mentioned in §4.3, it requires sample-wise ranking which incurs additional time. To study the relation between the time efficiency and performance, we change the $N$ used in reranking and plot the relation of $N$ and final accuracy in Figure 5, from which we observe that even with $N = 20$, RankRAG still improve the baseline model without reranking. While reranking across $N = 20$ to 100 improves the exact match score by 5.9% to 9.1% across three tasks, it incurs an additional $0.9\times$ to $6.0\times$ increase in time – *significantly less* than the $20\times$ to $100\times$ increase one might expect.

## 5.6 Case Study

Table 7 presents a case study on NQ dataset, where we observe that using retriever only yield noisy contexts, as there are several distractors, and some contexts (e.g. Passage 4/5 for ChatQA-1.5) are

---

[4] https://platform.openai.com/docs/api-reference/chat/create#chat-create-logprobs

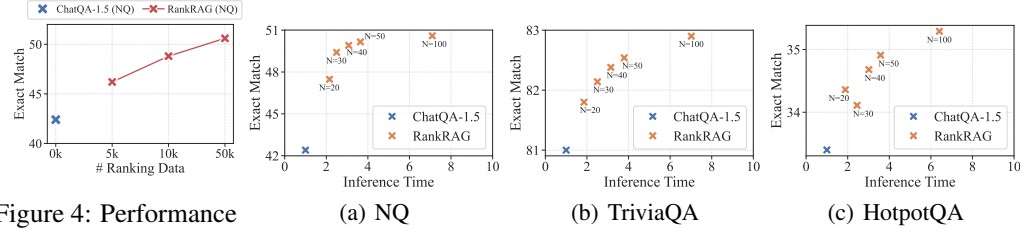

Figure 4: Performance w.r.t. # Ranking Data

(a) NQ  (b) TriviaQA  (c) HotpotQA

Figure 5: Performance v.s. Efficiency analysis for RankRAG.

Table 7: A case study on the top-retrieved context and predictions on NQ dataset, illustrating the effectiveness of RankRAG-8B over ChatQA-1.5-8B. Red text denotes distractors, while green stands for evidences. RankRAG is able to find the correct answer via extract more evidence with reranking.

**Q**: who hosted and won the inagural world cup? **A**: Uruguay

| | |
|---|---|
| ChatQA-1.5 | **Passage 1**: FIFA World Cup second round on home soil in 1982. England (1966) won its only title while playing as a host nation. Uruguay (1930), Italy (1934), Argentina (1978) and France (1998) won their first titles as host nations but have gone on to win again, while Germany (1974) won their second title on home soil...
**Passage 2**: FIFA World Cup hosts country is now chosen in a vote by FIFA's Congress ... Only Mexico, Italy, France, Germany (West Germany) until shortly after the 1990 World Cup) and Brazil have hosted the event on two occasions.
**Passage 3**: CONCACAF hosts, beating the bids of Canada and the United States, and thereby became the first nation to host two World Cups. This second World Cup in Mexico came 16 years after the first one in 1970...
**Passage 4**: 1998 FIFA World Cup Africa made their first appearances in the finals. France was awarded the 1998 World Cup on 2 July 1992 by the executive committee of FIFA during a general meeting in Zürich, Switzerland. They defeated Morocco by 12 votes to 7.
**Passage 5**: 2026 FIFA World Cup be hosted by one of the remaining four confederations: CONCACAF (North America; last hosted in 1994), CAF (Africa; last hosted in 2010), CONMEBOL (South America; last hosted in 2014), or OFC (Oceania, never hosted before)... **Prediction**: *Germany* (✘) |
| RankRAG | **Passage 1**: FIFA World Cup second round on home soil in 1982. England (1966) won its only title while playing as a host nation. Uruguay (1930), Italy (1934), Argentina (1978) and France (1998) won their first titles as host nations but have gone on to win again, while Germany (1974) won their second title on home soil...
**Passage 2**: Timeline of association football penalty kicks. Thirteen teams enter the first World Cup, held in Uruguay. The hosts beat Argentina 4–2 in the final. Contested between the top national teams of continental Europe, Dr. Gerö Cup' first edition is won by Italy.
**Passage 3**: The Uruguay national football team represents Uruguay in international association football and is controlled by the Uruguayan Football Association. They have won the Copa América 15 times, the most successful national team in the tournament, the most recent title being the 2011 edition. The team has won the FIFA World Cup twice, including the first World Cup in 1930 as hosts, defeating Argentina 4–2 in the final.
**Passage 4**: FIFA World Cup hosts country is now chosen in a vote by FIFA's Congress. The decision is currently made roughly seven years in advance of the tournament, though the hosts for the 2022 tournament were chosen at the same time as those for the 2018 tournament.
**Passage 5**: CONCACAF hosts, beating the bids of Canada and the United States, and thereby became the first nation to host two World Cups. This second World Cup in Mexico came 16 years after the first one in 1970... **Prediction**: *Uruguay* (✓) |

unhelpful. However, the utilization of reranking uncovers *two additional* relevant passages, aiding the model in providing the correct answer. More case studies are provided in Appendix G.

# 6 Conclusion

In this work, we introduce a new RAG framework, RankRAG, which instruction-tunes a single LLM for both ranking and answer generation. We find that the instruction tuned LLMs can outperform existing expert ranking models by only adding a small fraction of ranking data into the training blend. We compare our RankRAG with the state-of-the-art RAG models on comprehensive knowledge-intensive benchmarks and demonstrate RankRAG significantly outperform all of them on nine general-domain and five biomedical benchmarks for RAG.

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

# A    Dataset Description

The information for 14 datasets used in RankRAG is listed as follows.

## A.1    Main Experiments

- **NQ** (Kwiatkowski et al., 2019) is a widely used question-answering dataset constructed with Wikipedia. The questions are constructed from the Google search engine, and the answers are identified as text spans in the Wikipedia article.
- **TriviaQA** (Joshi et al., 2017) is a challenging QA dataset containing question-answer pairs from trivia enthusiasts and independently gathered evidence documents.
- **PopQA** (Mallen et al., 2023) is an entity-centric QA dataset concentrated on long-tail entities. For PopQA, we follow (Asai et al., 2024a) to use the long-tail subset, consisting of questions on 1399 rare entities whose monthly Wikipedia page views are less than 100.
- **HotpotQA** (Yang et al., 2018) is a multi-hop QA dataset, where the goal is to answer complex questions that require understanding and linking information from multiple documents.
- **2WikimQA** (Ho et al., 2020) is also a multi-hop QA designed to test machine understanding across two different Wikipedia entities, evaluating the ability of systems to handle cross-lingual and cross-cultural retrieval and question answering.
- **FEVER** (Thorne et al., 2018) is a fact verification dataset aimed at supporting research into the automatic verification of factual claims. It consists of claims that are manually verified against evidence from Wikipedia, providing a benchmark for fact-checking systems.
- **Doc2Dial** (Feng et al., 2020) is a document-grounded conversational QA dataset covering four domains: DMV, SSA, VA, and Student Aid. Each sample comprises a dialogue where a user poses queries regarding the document, and an agent responds those questions. The average document length is around 101K words.
- **TopiOCQA** (Adlakha et al., 2022) is grounded on the whole Wikipedia. It incorporates topic switching and requires the agent to search the entire Wikipedia for answers to user questions.
- **INSCIT** (Wu et al., 2023) is also grounded on the whole Wikipedia. It studies the case where user questions are under-specified and require clarification.

## A.2    Biomedical Benchmarks

- **MMLU-med** (Hendrycks et al., 2021) is a subset of six tasks related to biomedicine, including anatomy, clinical knowledge, professional medicine, human genetics, college medicine, and college biology. It contains 1089 questions in total.
- **MedQA** (Jin et al., 2021) is collected from the US Medical Licensing Examination, contaiing 1273 four-option multiple-choice questions focused on real-world scenarios from professional medical board exams.
- **MedMCQA** (Pal et al., 2022) includes multiple-choice questions derived from Indian medical entrance exams, covering 2400 healthcare topics across 21 medical subjects. We use the 4,183-question development set from MedMCQA, as the test set lacks provided ground truths.
- **PubmedQA** (Jin et al., 2019) is a biomedical research QA dataset consisting of 1000 manually annotated questions based on PubMed abstracts. Answers in PubMedQA are structured as yes/no/maybe to reflect the validity of the questions.
- **BioASQ** (Tsatsaronis et al., 2015) includes 618 questions constructed from biomedical literature without providing the ground truth snippets, challenging RAG systems to infer answers independently.

# B    Data Blending Details for Ranking-enhanced Instruction Finetuning

The dataset blending ratio for Stage-II is as follows:

- Drop: 0.069

- narrativeqa: 0.09
- quoref: 0.026
- ropes: 0.026
- Squad (Retrieval-augmented QA): 0.09
- Squad (Retrieval-augmented Ranking): 0.02
- WebQuestions (Retrieval-augmented QA): 0.09
- WebQuestions (Retrieval-augmented Ranking): 0.02
- newsqa: 0.09
- tatqa-arithmetic: 0.15
- tatqa-others: 0.08
- ConvQA: 0.2
- MS MARCO ranking: 0.15
- ConvQA ranking: 0.03
- SFT: 0.2

The ratio for each dataset is further normalized to ensure the total ratio equals to 1.

## C  Prompt Formats of Instruction Tuning

### C.1  Stage I: Supervised Fine-tuning

The format template of LLM inputs in stage-I is as follows:

```
System: This is a chat between a user and an artificial intelligence assistant.
The assistant gives helpful, detailed, and polite answers to the user's questions
based on the context. The assistant should also indicate when the answer cannot be
found in the context.

User: {Question 1}

Assistant: {Answer 1}

...

User: {Latest Question}

Assistant:
```

### C.2  Stage-II: Unified Instruction-Tuning for Ranking and Generation

The format template of LLM inputs in stage-II are as follows:

1) Context-rich QA data

```
System: This is a chat between a user and an artificial intelligence assistant.
The assistant gives helpful, detailed, and polite answers to the user's questions
based on the context. The assistant should also indicate when the answer cannot be
found in the context.

Passage: {(Gold) Passage containing relevant context for QA}

User: {Question 1}

Assistant: {Answer 1}
```

```
...

User: {Latest Question}

Assistant:
```

We tailor specific user instructions for various dataset types. For instance:

For datasets requiring short answers (such as DROP, NarrativeQA, Quoref, ROPES, SQuAD1.1, SQuAD2.0, NewsQA), we use: "Answer the following question with a short span."

For datasets that necessitate long answers (such as Synthetic_ConvQA), we instruct: "Please give a full and complete answer for the question."

For datasets involving arithmetic calculations or number extraction from the context (such as TAT-QA), we specify: "Answer the following question with a number from the context or through math arithmetic."

For datasets that may require both short and long answers (such as TAT-QA-Others), we direct: "Answer the following question with a short span, or a full and complete answer."

2) Retrieval-augmented QA data

```
System: This is a chat between a user and an artificial intelligence assistant.
The assistant gives helpful, detailed, and polite answers to the user's questions
based on the context. The assistant should also indicate when the answer cannot be
found in the context.

Passage 1: {(Shuffled) Passage 1}

Passage 2: {(Shuffled) Passage 2}

Passage 3: {(Shuffled) Passage 3}

Passage 4: {(Shuffled) Passage 4}

Passage 5: {(Shuffled) Passage 5}

...

User: {Question}

Assistant:
```

3) Context ranking data

```
System: This is a chat between a user and an artificial intelligence assistant.
The assistant gives helpful, detailed, and polite answers to the user's questions
based on the context. The assistant should also indicate when the answer cannot be
found in the context.

Passage: {Passage 1}

User: {For the question <question>, access whether the passage is relevant to the
question. Return True if relevant, otherwise False. }

Assistant:
```

4) Retrieval-augmented ranking data

```
System: This is a chat between a user and an artificial intelligence assistant.
The assistant gives helpful, detailed, and polite answers to the user's questions
based on the context. The assistant should also indicate when the answer cannot be
```

```
found in the context.

Passage 1: {(Shuffled) Passage 1}

Passage 2: {(Shuffled) Passage 2}

Passage 3: {(Shuffled) Passage 3}

Passage 4: {(Shuffled) Passage 4}

Passage 5: {(Shuffled) Passage 5}

User: {For the question <question>, access whether the above passages are relevant
to the question. Return all the relevant passage id. }

Assistant:
```

## D    Prompt Formats of Target Tasks

### D.1    Context Ranking

NQ/TriviaQA/HotpotQA/PopQA:

```
System: This is a chat between a user and an artificial intelligence assistant.
The assistant gives helpful, detailed, and polite answers to the user's questions
based on the context. The assistant should also indicate when the answer cannot be
found in the context.

Passage: {Passage}

User: {For the question <question>, access whether the passage is relevant to the
question. Return True if relevant, otherwise False. }

Assistant:
```

FEVER:

```
System: This is a chat between a user and an artificial intelligence assistant.
The assistant gives helpful, detailed, and polite answers to the user's questions
based on the context. The assistant should also indicate when the answer cannot be
found in the context.

Passage: {Passage}

User: {For the claim <claim>, access whether the passage is relevant to the
claim. Return True if relevant, otherwise False. }

Assistant:
```

Doc2dial, Inscit, TopiocQA:

```
System: This is a chat between a user and an artificial intelligence assistant.
The assistant gives helpful, detailed, and polite answers to the user's questions
based on the context. The assistant should also indicate when the answer cannot be
found in the context.

Passage: {Passage}

User: {Question 1}
```

```
Assistant: {Answer 1}

...

User: {For the question <latest question>, access whether the passage is relevant
to the question. Return True if relevant, otherwise False. }

Assistant:
```

## D.2  RAG

NQ/TriviaQA/HotpotQA/PopQA:

```
System: This is a chat between a user and an artificial intelligence assistant.
The assistant gives helpful, detailed, and polite answers to the user's questions
based on the context. The assistant should also indicate when the answer cannot be
found in the context.

Passage 1: {Rerank Top Passage 1}

Passage 2: {Rerank Top Passage 2}

Passage 3: {Rerank Top Passage 3}

Passage 4: {Rerank Top Passage 4}

Passage 5: {Rerank Top Passage 5}

...

User: {Question}. Answer the above question with a short phrase.

Assistant:
```

Fever:

```
System: This is a chat between a user and an artificial intelligence assistant.
The assistant gives helpful, detailed, and polite answers to the user's questions
based on the context. The assistant should also indicate when the answer cannot be
found in the context.

Passage 1: {Rerank Top Passage 1}

Passage 2: {Rerank Top Passage 2}

Passage 3: {Rerank Top Passage 3}

Passage 4: {Rerank Top Passage 4}

Passage 5: {Rerank Top Passage 5}

...

User: Answer the following question with True or False. Is the claim '<claim>' correct?

Assistant:
```

Doc2dial, Inscit, TopiOCQA:

```
System: This is a chat between a user and an artificial intelligence assistant.
```

```
The assistant gives helpful, detailed, and polite answers to the user's questions
based on the context. The assistant should also indicate when the answer cannot be
found in the context.

Passage 1: {Rerank Top Passage 1}

Passage 2: {Rerank Top Passage 2}

Passage 3: {Rerank Top Passage 3}

Passage 4: {Rerank Top Passage 4}

Passage 5: {Rerank Top Passage 5}

User: {Question 1}

Assistant: {Answer 1}

...
User: {Latest Question}

Assistant:
```

# E  Additional Experiment Results

## E.1  Ranking Performance Using DPR and Contriever as Retrievers $\mathcal{R}$

Table 8 shows the ranking performance of RankRAG-8B using DPR (Karpukhin et al., 2020) and
Contriever (Izacard et al., 2022) on three datasets. There are consistent performance gains for all
tasks, indicating that RankRAG can apply to many popular retrieval models to improve the quality of
retrieved contents.

Table 8: Answer Recall Comparison Before and After Ranking on 3 Representative Datasets.

| NQ | DPR | | | Contriever | | |
|---|---|---|---|---|---|---|
| | R@5 | R@10 | R@20 | R@5 | R@10 | R@20 |
| Before Ranking | 69.50% | 76.20% | 81.00% | 67.60% | 75.24% | 80.67% |
| w/ RankRAG | 77.95% | 81.70% | 84.56% | 75.32% | 80.18% | 84.70% |
| **TriviaQA** | **DPR** | | | **Contriever** | | |
| | R@5 | R@10 | R@20 | R@5 | R@10 | R@20 |
| Before Ranking | 67.80% | 74.20% | 80.30% | 81.95% | 86.76% | 90.08% |
| w/ RankRAG | 77.73% | 79.40% | 84.74% | 88.71% | 90.05% | 92.59% |
| **PopQA** | **DPR** | | | **Contriever** | | |
| | R@5 | R@10 | R@20 | R@5 | R@10 | R@20 |
| Before Ranking | 43.60% | 48.90% | 54.25% | 60.61% | 65.54% | 69.90% |
| w/ RankRAG | 50.32% | 53.75% | 57.76% | 65.11% | 68.41% | 71.77% |

## E.2  RAG Performance with Different $k$

We also show the performance of RankRAG with different context size $k$ in figure 6. From the result,
we observe that different from the trend of vanilla RAG approaches (without ranking), $k = 5$ already
works well for most datasets. This effectiveness stems from the reranking step, which prioritizes the
most relevant contexts at the top, reducing the necessity to include additional contexts.

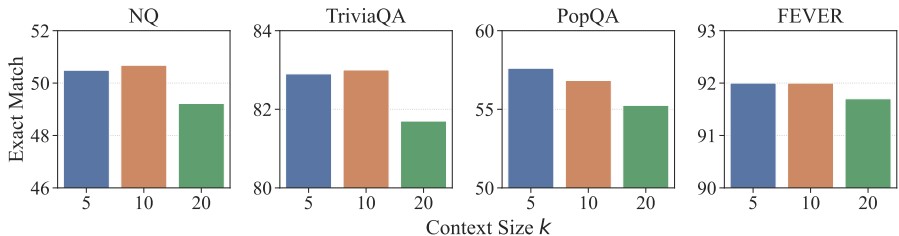

Figure 6: Performance of RankRAG on different context size $k$.

# F  Performance of NQ and Trivia QA on DPR Splits

We observe that the NQ and TriviaQA datasets exist in two versions: one used by the DPR (Karpukhin et al., 2020) and FiD (Izacard & Grave, 2021) papers, which include 3610 and 11316 questions for NQ and TriviaQA, respectively. In contrast, the KILT benchmark (Petroni et al., 2021) utilizes only subsets of these, comprising 2837 and 5355 examples for NQ and TriviaQA, respectively. It is noteworthy that many recent studies report performance metrics on these datasets without clarifying which version was employed for evaluation.

To facilitate an *honest* and *fair* comparison, we present the performance of RankRAG on both datasets using the DPR splits in Table 9. Notably, regardless of the subset used, RankRAG consistently outperforms both ChatQA and Llama-3-instruct, our direct competitors, as well as other methods utilizing InstructGPT as backbones. We aim for these results to assist the community in making accurate comparisons when referring to the performance of RankRAG.

Table 9: Performance Across Models.

| Model | Model Configuration | NQ EM (%) | TriviaQA EM / Acc. (%) |
|---|---|---|---|
| *Representative Baselines* | | | |
| OpenAI GPT | GPT-3.5-0613 | 35.2 | 70.1 / 81.3 |
| | GPT-3.5-0613 RAG | 42.3 | 65.8 / 76.7 |
| | GPT-4-0613 | 37.2 | **72.6 / 85.1** |
| | GPT-4-0613 RAG | 36.2 | 61.2 / 75.9 |
| | GPT-4-turbo-2024-0409 | 38.3 | 68.0 / 84.5 |
| | GPT-4-turbo-2024-0409 RAG | 36.3 | 57.6 / 79.2 |
| *Using Llama-2 (Touvron et al., 2023) as the backbone LLM* | | | |
| Llama-2-Chat | Llama-2 RAG 70B | 37.7 | 65.6 / – |
| ChatQA-1.0 | Llama-2 7B | 37.0 | 62.4 / 74.3 |
| | Llama-2 13B | 43.9 | 66.6 / 76.9 |
| | Llama-2 70B | 45.0 | 69.8 / 80.2 |
| RankRAG | Llama-2 7B | 42.4 | 68.3 / 78.9 |
| | Llama-2 13B | 46.2 | 69.5 / 80.0 |
| | Llama-2 70B | 48.7 | 72.3 / 82.6 |
| *Using Llama-3 (Meta-AI, 2024) as the backbone LLM* | | | |
| Llama-3-Instruct | Llama-3-Instruct RAG 8B | 27.6 | 57.1 / 74.6 |
| | Llama-3-Instruct RAG 70B | 37.3 | 67.6 / 79.6 |
| ChatQA-1.5 | Llama-3 8B | 44.1 | 65.4 / 75.8 |
| | Llama-3 70B | 46.0 | 69.0 / 80.4 |
| RankRAG | Llama-3 8B | 46.1 | 68.8 / 79.9 |
| | Llama-3 70B | **50.0** | **72.6** / 82.9 |

# G  Additional Case Studies

Tables 10 and 11 provide additional examples from the PopQA and HotpotQA datasets, which focus on long-tailed and multi-hop QA. These tasks are particularly challenging for retrievers, making it difficult to obtain relevant context from the corpus. Consequently, ChatQA-1.5 often struggles to produce the correct answers. However, the reranking step in RankRAG helps counteract poor initial

Table 10: A case study on the top-retrieved context and predictions on PopQA dataset, illustrating the effectiveness of RankRAG-8B over ChatQA-1.5-8B. Red text denotes distractors, while green stands for evidences.

| **Q**: Who is the author of The Universe Around Us? | **A**: James Hopwood Jeans / James Jeans / Sir James Jeans / Sir James Hopwood Jeans |
|---|---|
| ChatQA-1.5 | **Passage 1**: Hooper is the author of two books published by Smithsonian Books/HarperCollins. The first, Dark Cosmos: In Search of our Universe's Missing Mass and Energy (2006) was named a notable book by Seed Magazine. His second book, Nature's Blueprint: Supersymmetry and the Search for a Unified Theory of Matter and Force (2008), was called "essential reading" by New Scientist... **Passage 2**: Fraknoi is the author or co-author of 14 books in the field of astronomy. He was the lead author of Voyages through the Universe, an introductory college astronomy textbook published by Brooks-Cole, which went through three editions. In the 1980s, he co-edited with Byron Preiss two collections of science articles and science fiction stories, "The Universe" and "The Planets." With Sidney Wolff, Fraknoi founded and was co-editor of the first on-line journal devoted to astronomy education, "Astronomy Education Review"... **Passage 3**: The Universe" and "The Planets." With Sidney Wolff, Fraknoi founded and was co-editor of the first on-line journal devoted to astronomy education, "Astronomy Education Review". He edited two collections of resources for K-12 teachers, The Universe at Your Fingertips and More Universe at Your Fingertips published through the Astronomical Society of... **Passage 4**: Lincoln Kinnear Barnett (1909–1979) was an editor and author, most notably at Life Magazine for many years. Lincoln Barnett wrote a number of books, including "The Universe and Doctor Einstein" **Passage 5**: The Universe Maker is a science fiction novel by American author A.E. van Vogt, published in 1953 by Ace Books as an Ace Double with The World of Null-A. It is based on the authors ́ "The Shadow Men" (Startling Stories, 1950). Set 400 years into the future, the main character is Morton Cargill, a U.S. Army officer who served in the Korean War... 
 **Prediction**: Lincoln Barnett (✘) |
| RankRAG | **Passage 1**: The Universe Around Us is a science book written by English astrophysicist Sir James Jeans, first published in 1929 by the Syndics of the Cambridge University Press.... **Passage 2**: These books made Jeans fairly well known as an expositor of the revolutionary scientific discoveries of his day, especially in relativity and physical cosmology. In 1939, the Journal of the British Astronomical Association reported that Jeans was going to stand as a candidate for **Passage 3**: James Jeans books for the lay public, including "The Stars in Their Courses" (1931), "The Universe Around Us", "Through Space and Time" (1934), "The New Background of Science" (1933), and "The Mysterious Universe. **Passage 4**: The Universe Around Us no special scientific knowledge. Parts of the book cover the same ground as various lectures I have recently delivered to University and other audiences, including a course of wireless talks I gave last autumn. It has been found necessary to rewrite these almost in their entirety, so that very few sentences remain in their original form, but those who have asked me to publish my lectures and wireless talks will find the substance of them in the present book. **Passage 5**: Lincoln Barnett Lincoln Kinnear Barnett (1909–1979) was an editor and author, most notably at Life Magazine for many years. Lincoln Barnett wrote a number of books, including "The Universe and Doctor Einstein", "The World We Live In", and "The Treasure of Our Tongue". The Universe and Doctor Einstein is a layman's introduction to the theory of relativity. It includes a foreword by Albert Einstein, and has been reprinted several times... 
 **Prediction**: Sir James Jeans (✔) |

Table 11: A case study on the top-retrieved context and predictions on HotpotQA dataset, illustrating the effectiveness of RankRAG-8B over ChatQA-1.5-8B. Red text denotes distractors, while green stands for evidences.

| **Q**: Which song did Eminem and Rihanna collaborate on after their other collaboration song in studio album "Unapologetic?" | **A**: The Monster |
|---|---|
| ChatQA-1.5 | **Passage 1**: Unapologetic is the seventh studio album by Barbadian singer Rihanna. It was released on November 19, 2012, by Def Jam Recordings and SRP Records. It was recorded between June and November 2012, during promotion of her sixth album, "Talk That Talk" (2011). As executive producer, Rihanna enlisted previous collaborators The-Dream, David Guetta, Chase & Status, and StarGate to work alongside new collaborators such as Parker Ighile, Mike Will Made-It, and Labrinth... **Passage 2**: Def Jam France announced via Twitter that Rihanna would release a new single the upcoming week while her seventh studio album was scheduled to be released in November 2012. On October 11, 2012, in one of her tweets revealed that the title of her new album is "Unapologetic" alongside with its cover. "What Now" was written by British singer-songwriter Livvi Franc together with Rihanna, Parker Ighile and Nathan Cassells, while the production of the song was done by the latter two. Ighile and Cassells... **Passage 3**: Justin then went on to co write "Stay" with Mikky Ekko and recorded by Barbadian singer Rihanna for her seventh studio album, "Unapologetic" (2012). It features guest vocals by Mikky Ekko, and was released as the second single from the album on 7 January 2013. The song reached the top five of twenty-four countries worldwide including number four in the UK and number three on the US Billboard Hot 100, becoming Rihanna's twenty-fourth top ten on the latter chart... **Passage 4**: Via her official Twitter account, Rihanna posted series of "teasing" tweets announcing her seventh studio album. On October 11, 2012, in one of her tweets revealed that the title of her new album is "Unapologetic" alongside with its cover. "Jump" is the overall seventh and final single off Unapologetic. It was written by Kevin Cossom and M. B. Williams together with its producers StarGate (Mikkel S. Eriksen and Tor Erik Hermansen) and Chase & Status (Saul Milton **Passage 5**: copies of the song were sold in the UK, making "Love the Way You Lie" the country's biggest-selling song of 2010. The same year, a sequel to the single, titled "Love the Way You Lie (Part II)" was released as part of Rihanna's fifth studio album "Loud"; it mainly views matters from the female protagonist perspective. In November 2012, Eminem and Rihanna collaborated again on "Numb", which was included on Rihanna's seventh album "Unapologetic"... 
 **Prediction**: Love the Way You Lie (✘) |
| RankRAG | **Passage 1**: The Monster (song). Copies of the song were sold in the UK, making "Love the Way You Lie" the country's biggest-selling song of 2010. The same year, a sequel to the single, titled "Love the Way You Lie (Part II)" was released as part of Rihanna's fifth studio album "Loud"; it mainly views matters from the female protagonist perspective. In November 2012, Eminem and Rihanna collaborated again on "Numb", which was included on Rihanna's seventh album "Unapologetic".... **Passage 2**: "Numb" is a song by Barbadian singer Rihanna from her seventh studio album "Unapologetic" (2012). It features guest vocals by American rapper Eminem, making it the pair's third collaboration since the two official versions of "Love the Way You Lie". Following the album's release, "Numb" charted on multiple charts worldwide including in Canada, the United Kingdom and the United States. "Numb" lasts for a duration of . **Passage 3**: Eminem also wanted to experiment with "retro, vintage" sounds such as beatbreaks and scratches, and he felt that Rubin could help him "take that to another level." Rihanna, with whom Eminem previously collaborated on "Love the Way You Lie" from Eminem's previous studio effort, "Recovery" (2010), was featured on the song "The Monster". On September 11, 2013, she hinted at the... **Passage 4**: together with Jay-Z, Bono and The Edge for the same campaign to alleviate the 2010 Haiti earthquake. In summer 2010, Rihanna collaborated with rapper Eminem on "Love the Way You Lie", which was a major worldwide success, reaching No. 1 in over 20 countries. Reaching number 2, the song became the biggest-selling song of 2010 in the UK and the first of Rihanna's singles to sell over a million copies in the country. In October 2010, Rihanna switched managers ... **Passage 5**: Eminem asked for more tracks and subsequently heard "Love the Way You Lie". He chose it and told his manager Paul Rosenberg he wanted to collaborate with the Barbadian singer Rihanna. Eminem told Skyrock, "It's one of those tracks that I felt like only she could pull it off." Rosenberg sent the track to Rihanna, who accepted Eminem's request "at the last moment." Eminem then wrote the rapped verses. 
 **Prediction**: The Monster (✔) |

retrieval by finding more pertinent evidence. Coupled with RAG-oriented finetuning, RankRAG effectively filters out distracting entities and pinpoints the correct answers.

