# OpenReview forum: "RankRAG: Unifying Context Ranking with Retrieval-Augmented Generation in LLMs"
_NeurIPS.cc/2024/Conference — NeurIPS 2024 poster_

### Official Review · Reviewer_y5n5 · 2024-07-08

**Soundness:** 2
**Presentation:** 3
**Contribution:** 2
**Rating:** 5
**Confidence:** 4

**Summary:**

This paper proposed a method about how to use a single LLM for both context reranking and answer in RAG tasks. Particularly, they finetuned a LLM with both ranking data and QA data with two stages. For inference, they use the trained llm to sample the top_k contexts at first, and then input them into llm to get the answer. Compared with normal rag workflow, this method choose the contexts by llm itself. Experiments in some QA tasks evaluated the QA effectiveness of this LLM.

**Strengths:**

1. Two stages of training enhance the reranking capacity of llm using fewer ranking data.
2. Using just a single llm for context ranking and answer at the same time.

**Weaknesses:**

1. The motivation is unclear. In lines 23 to 31, limitations 1 and 2 precisely explain the need for a reranker, which is also a challenge related to retrieval. Only with limited text in limitation 3  briefly mentions why the LLM is used for reranking contexts due to zero-shot performance, which has been proved with other rerankers in this paper. It is unclear about why using llm reranking itself rather than a separate reranker, maybe including the semantic gap between the reranker and llm or the larger length of reranking input at once.
2. Insufficient experiments in same reranking setting. It is lack of baselines with RAG methods also using a reranker, which can prove the core reranking effectiveness of RankRAG, such as rankgpt(Is ChatGPT Good at Search? Investigating Large Language Models as Re-Ranking Agents) and rankvicuna(RankVicuna: Zero-Shot Listwise Document Reranking with Open-Source Large Language Models). And also, the amount of top-k contexts after reranking is also hard to find in Table 2&6. How many contexts for reranking are input into llm at once? In addition, there are some blank spaces in Table 2 that are not convincing.
3. Incremental techniques. Compared with ChatQA, RankRAG introduced the new training data RQA and RAR in stage 2th and reranking phrase during inference. But as shown in ablation study, each component contribute a little.

**Questions:**

1. Can you analyze the relationship between high retrieval call and QA performance in RankRAG for limitation2 on line 26? Can you provide further analysis to demonstrate that your method can solve this limitation?

2. Have you conducted experiments to test whether RankRAG's reranking performance is robust when using different retrievers on lines 300?

**Limitations:**

No potential negative societal impacts.

---

> ### Author Rebuttal · Authors · 2024-08-07
>
> Many thanks for your comments and feedback. We discuss your raised points in the following.
>
> ---
> > “1. The motivation is unclear. … It is unclear about why using llm reranking itself rather than a separate reranker, maybe including the semantic gap between the reranker and llm or the larger length of reranking input at once.”
> - Thank you for this insightful comment. The key advantages of using an LLM for both ranking and generation include:
>
> - **Performance**: As shown in Table 6 of the main paper, the reranking performance with LLMs surpasses that of state-of-the-art ranking models, likely due to enhanced transfer learning between RAG and ranking tasks facilitated by similar input formats (Table 1).
>
> - **Label Efficiency**: LLM-based reranking reduces the need for labeled data, as demonstrated by RankRAG's superiority over RankLlama, which requires ten times more data. This is particularly beneficial in resource-scarce settings.
>
> - **Memory Efficiency**: RankRAG operates with a single model during inference, enhancing deployment efficiency.
> To further illustrate the performance gains, we have exhibited the RAG performance of ChatQA-1.5, Llama-3-instruct, and RankRAG using RankLlama as the Reranker in Table 4 of the supplementary PDF. The results justify that RankRAG outperforms these on 8 of 9 datasets, with significant gains on 6 datasets.
>
> ---
> > “2. Insufficient experiments in same reranking setting. It is lack of baselines with RAG methods also using a reranker, which can prove the core reranking effectiveness of RankRAG, such as rankgpt(Is ChatGPT Good at Search? Investigating Large Language Models as Re-Ranking Agents) and rankvicuna(RankVicuna: Zero-Shot Listwise Document Reranking with Open-Source Large Language Models)”
>
> - Thanks for pointing out these works. We have compared RankRAG with RankVicuna, RankGPT-3.5 and RankGPT-4 in Table 6 of the supplementary PDF.  We observe that RankRAG outperforms all of them on 5 datasets. We will mention these papers in the “related work” section and add these comparison in the next version of the work.
>
> ---
> > “And also, the amount of top-k contexts after reranking is also hard to find in Table 2&6. How many contexts for reranking are input into llm at once? ”
> - We set a fixed k=5 for RankRAG to simplify hyperparameter tuning and expedite inference. The performance of RankRAG across different k is in Figure 6, Appendix G.2.
>
> ---
> > “In addition, there are some blank spaces in Table 2 that are not convincing.”
>
> - Thanks for pointing this out. In table 2,  most numbers are from existing papers to ensure fair comparisons and many existing models are either not-public (e.g. PaLM2, RA-DIT, RePlug) or only test on a small number of tasks, thus, it is somehow challenging for us to provide additional results for these models. In response to your suggestion, we included results for all 9 datasets using three OpenAI models (GPT-3.5-turbo-1106, GPT-4-0125, GPT-4-turbo-0409), both with and without retrieval, detailed in Table 5 of the supplementary PDF. RankRAG outperforms these models on 7 out of 9 datasets, achieving average gains of 5.8% and 9.3% for the 8B and 70B variants, respectively.
>
> ---
> > “3. Incremental techniques. Compared with ChatQA, RankRAG introduced the new training data RQA and RAR in stage 2th and reranking phrase during inference. But as shown in ablation study, each component contribute a little.”
>
> - With all these techniques combined, our RankRAG-8B consistently outperforms ChatQA-1.5-8B on all 9 RAG tasks. RQA and RAR contribute to this performance gain in the majority of these tasks—6 for RQA and 8 for RAR, respectively. Notably, the final RankRAG-8B even outperforms much larger ChatQA-1.5-70B on NQ (50.6 vs. 47.0), PopQA (57.7 vs. 50.9), and Inscit (33.3 vs. 32.3). Its average score across nine datasets (52.6) is 3 points higher than same size ChatQA-1.5-8B (49.6), 5.5 points higher than much larger Llama3-Instruct 70B (47.1), while being just 1 point below SOTA ChatQA-1.5-70B (53.6). It also outperforms other  state-of-the-art large models including Mixtral-8x22B-Instruct and RePlug 65B.
>
> - Our RankRAG-70B outperforms ChatQA-1.5-70B on 8 out of 9 RAG tasks. Its average score 56.1 is also much better than ChatQA-1.5-70B’s 53.6, which already suppasses GPT-4 Turbo on many RAG tasks. In this work, we improve the performance of frontier-class ChatQA-1.5 by a margin, which is non-trivial.
>
> ---
> > “ Questions: 1. Can you analyze the relationship between high retrieval call and QA performance in RankRAG for limitation2 on line 26? Can you provide further analysis to demonstrate that your method can solve this limitation?”
> - Table 6 highlights that the initial recall of Dragon retriever is often inadequate, with Recall@5 below 70% for PopQA and under 50% for HotpotQA. RankRAG significantly boosts the recall of relevant content, improving Recall@5 by 12% and 9% for PopQA and HotpotQA, respectively. This enhancement in recall translates into notable performance improvements—Table 4 shows absolute gains of 8% and 4% on these two datasets. These findings further justify RankRAG's effectiveness in addressing these limitations.
>
> ---
> > “ Questions: 2. Have you conducted experiments to test whether RankRAG's reranking performance is robust when using different retrievers on lines 300?”
> - Yes, we detailed the reranking performance in Table 8, Appendix G.1 of our manuscript. There, we find that RankRAG's reranking notably enhances performance, achieving over **8% gains for DPR** and **7% for Contriever** in terms of Recall@5 on average, compared to using only the retriever.
>
> ---
> ---
> Thank you once again for your insightful review. We appreciate your feedback on our work. Please let us know if you have any further questions, and we are happy to discuss further.

---

> ### Author Response · Authors · 2024-08-13
> **A Gentle Reminder**
>
> Dear Reviewer y5n5,
>
> Thank you again for your detailed comments and constructive suggestions. We will incorporate all of them into the final version of our submission. We hope our response can help address your concerns. As the discussion period is closing, please let us know if you have any further questions. We would be happy to discuss them further with you.
>
> Best,
>
> Authors

---

### Official Review · Reviewer_LbZs · 2024-07-13

**Soundness:** 4
**Presentation:** 4
**Contribution:** 4
**Rating:** 8
**Confidence:** 4

**Summary:**

The authors introduce a novel approach to instruction fine-tuning large language models (LLMs) for ranking and answer generation tasks.
Their approach involves two main steps:
1. Supervised Fine-Tuning: Initially, the LLM is fine-tuned on a general instruction-following dataset.
2. Ranking Task Fine-Tuning: The LLM is then further fine-tuned using a mix of different instruction ranking datasets that contains multiple ranking-oriented tasks.

Incorporating ranking-based data during fine-tuning enhances the LLM's ability to rank documents retrieved through retrieval-augmented generation (RAG). The fine-tuned LLM is then used to rank the RAG-retrieved documents, and only the top-k ranked documents are added in the context.

The authors demonstrate that their method significantly improves the LLM's performance for knowledge intensive RAG based tasks.

**Strengths:**

1. The authors introduce a novel method of fine-tuning LLMs for combined ranking and answer generation tasks.
2. The proposed method outperforms existing approaches on RAG benchmarks, especially on challenging datasets where RAG retrieval is suboptimal, due to the LLM-based re-ranking.
3. Their model exceeds the performance of re-ranking models trained on much larger datasets.
4. The fine-tuned LLM demonstrates strong generalization capabilities, showcased by its performance on medical benchmarks.
5. The paper is well-written and includes exhaustive experiments.

**Weaknesses:**

1. Scoring each document individually increases the latency significantly. Have you considered scoring multiple documents simultaneously? Similar to retrieval-augmented ranking dataset, you could input a group of documents and score them in a single pass. The group size could be tunable, balancing performance drop against latency improvement.
2. It would be helpful to include examples where RankRAG fails, particularly in cases where relevant documents are not ranked higher. Providing these examples can help understand scenarios where it might not perform optimally.

**Questions:**

N/A

**Limitations:**

The authors have acknowledged the limitations, specifically in terms of latency and its lack of training on code or mathematical data.

---

> ### Author Rebuttal · Authors · 2024-08-07
>
> Many thanks for your comments and feedback. We discuss your raised points in the following.
>
> ---
> > "1. Scoring each document individually increases the latency significantly. Have you considered scoring multiple documents simultaneously? Similar to retrieval-augmented ranking dataset, you could input a group of documents and score them in a single pass. The group size could be tunable, balancing performance drop against latency improvement."
> - Thank you for this nice suggestion. In our initial attempt, we tried to use the listwise reranking approach with a similar format to the retrieval-augmented ranking dataset to fulfill the reranking purpose but did not observe the performance gain. We will definitely consider your suggestion to explore how to further improve the efficiency of the reranking step.
>
> ---
> > "2. It would be helpful to include examples where RankRAG fails, particularly in cases where relevant documents are not ranked higher. Providing these examples can help understand scenarios where it might not perform optimally."
>
> - Thank you for this excellent suggestion. We will include the examples in the final version of the paper. Specifically, RankRAG may encounter difficulties in the following scenarios:
>   - QA involving long-tailed knowledge, where poor initial retrieval excludes relevant documents from the top-N contexts, preventing further ranking.
>   - Multi-hop QA tasks, where finding relevant documents can be challenging as it requires multi-hop reasoning beyond simple keyword or semantic matching.
> To alleviate these issues, potential solutions include incorporating more powerful retrieval models [1] or implementing a multi-step reranking strategy [2].
>
> [1] Wang et al. "Improving text embeddings with large language models." arXiv preprint arXiv:2401.00368 (2023).
>
> [2] Khalifa et al. "Few-shot reranking for multi-hop QA via language model prompting." arXiv preprint arXiv:2205.12650 (2022).
>
> ---
> ---
> Thank you once again for your insightful review. We appreciate your feedback on our work. If you have any further questions, please let us know. We would be happy to discuss them further.

---

> > ### Comment · Reviewer_LbZs · 2024-08-13
> >
> > Thanks for the response. Including failure cases in the final version would be great.

---

> > > ### Author Response · Authors · 2024-08-13
> > >
> > > Thank you again for your insightful feedback! We will incorporate them in our next version of the manuscript.

---

### Official Review · Reviewer_ze6x · 2024-07-19

**Soundness:** 4
**Presentation:** 3
**Contribution:** 3
**Rating:** 8
**Confidence:** 5

**Summary:**

This work proposes an effective approach that enhances existing RAG methods by introducing an additional context refinement step. This step filters out retrieved, but non-relevant contexts prior to including them as context in the input for answer generation.

The authors train context reranking alongside answer generation using a single LLM. They demonstrate that adding only a fraction of ranking data to the training blend yields superior ranking performance compared to training the same model in isolation on the ranking data, while outperforming other models that have been trained on 10 times more ranking data.

For evaluation, the authors compare their method on 9 general- and 5 specific-domain datasets, showing consistent improvements across LLM sizes for the Llama model family (Llama 2 and 3) over existing methods.

**Strengths:**

- The paper introduces the novel idea of using the same LLM to first assess the relevance of individual contexts in a cross-encoder ranking style before using them as input for answer generation.

- The proposed RankRAG outperforms existing methods on various general and specific-domain datasets.

- Extensive experimentation and ablations covers a wide range of possible setups including LLM size, retriever, and efficiency and effect of different model components.

**Weaknesses:**

Main concern:
- Reranking contributes only around 5% of the overall effectiveness on average (Table 4 RankRAG compared to Llama3-ChatQA-1.5-X), and the 7x computational overhead in inference time raises questions about its realistic application, given the computational demands of large models already without ranking contexts. The ratio of performance gained ( in combination with no sig. testing) and increase in computation is my main issue with this work. I am aware reranking fewer contexts decreases performance, however, similarity decreases performance gain over other methods.

- No significance testing was done (for both generation on ranking ) to strengthen effectiveness claims as differences for most datasets are minor. Authors
justify in their additionally provided checklist that sig. testing is not needed as generation and ranking is deterministic, this however, touches upon a different aspect and does not remove the need for testing whether the performance is significantly better than previous methods.
Further, improvements in Table 2 stem from NQ and PopQA which are relatively simple datasets that can be answered with a single context, therefore it is not apparent why RankRAG would particularly excel at those. Moreover, the ranking performance for these datasets in Table 6 only marginally improves over other rerankers, therefore it is to be expected to obtain similar gains when replacing the RankRAG reranker module  with other strong rerankers. As a side note averaging over different metrics - even though seen in many recent works - should not be done.


Other points to improve upon:
- No information about the crowdsourced dataset is provided.
- In Section 4.2.3, the claim that LLM needs to be robust against irrelevant context contradicts the proposal to filter out irrelevant context beforehand.

Some experimental setups are unclear:
- It is not described how true/false tokens from context ranking are translated into a score that can be used for ranking.
- In Section 5.1, it is not clear if baselines use different retrieval setups; otherwise, effectiveness claims do not seem valid.
- It is unclear which number k is eventually used in RankRAG. The paper mentions optimizing for
k=5,10,20, the baselines.

- The related work section could mention GRITLM as the first model jointly training answer generation and ranking for RAG: Muennighoff, Niklas, et al. "Generative representational instruction tuning." arXiv preprint arXiv:2402.09906 (2024).

Issues in Writing:

- Line 332: "Observe that even with N = 20, We noted that" – incomplete sentence.

- Line 41: "in RAG framework" -> "in the RAG framework".

- Caption Fig 1: "the strongest RAG model," -> "the strongest RAG models,".

- Line 100: "embedding model" -> "embedding models".

- Line 111: "As illurstraded" -> "As illustrated".

- Line 112: "one of the strongest model." -> "one of the strongest models".

- Line 157: "that, it is" -> "that it is".

- Line 165: "The LLM need" -> "The LLM needs".

- Line 169: "ranking data are" -> "ranking data is".

**Questions:**

- Why is a fixed number of contexts used? Some questions might need more contexts such as multi-hop datasets while others such as NQ (single context dataset) would need less. Instead of a score cutoff, why not use a dynamic number of contexts that is determined by the binary true/false label the mode already outputs?
- In Section 5.4, the paper states that it incurs an additional 1.8× to 7.0× increase in time, significantly less than the 20× to 100× increase one might expect. It is not sufficiently explained why one would expect a 100x increase in time.
- The approach for reranking context looks at passages individually, but retrieval training included listwise ranking. Was listwise ranking also tried for context reranking?

**Limitations:**

The authors addressed the limitations adequately, however, could emphasize more on the dramatic inference time increase that results from the reranking step.

---

> ### Author Rebuttal · Authors · 2024-08-07
>
> Many thanks for your comments and feedback. We discuss your main concerns in the following.
>
> ---
> > “Reranking contributes only around 5% of the overall effectiveness on average (Table 4 RankRAG compared to Llama3-ChatQA-1.5-X), and the 7x computational overhead in inference time raises questions about its realistic application
> - First, we would like to clarify that the 7x computational overhead of ChatQA-1.5-8B occurs when RankRAG-8B reranks 100 retrieved contexts. By doing so, the smaller RankRAG-8B outperforms most state-of-the-art large models including Mixtral-8x22B-Instruct and RePlug 65B, except ChatQA-1.5 70B and Claude 2. Notably, RankRAG-8B even outperforms ChatQA-1.5-70B on NQ (50.6 vs. 47.0), PopQA (57.7 vs. 50.9), and Inscit (33.3 vs. 32.3). Its average score across nine datasets (52.6) is 3 points higher than same size ChatQA-1.5-8B (49.6), 5.5 points higher than much larger Llama3-Instruct 70B (47.1), while being just 1 point below SOTA ChatQA-1.5-70B (53.6).  These accuracy gains over frontier-class models are not marginal.
>
> - Second, RankRAG 70b only uses top-30 contexts for reranking (mentioned in Line 242) with around 2.5x overhead when compared to ChatQA-70b.
>
> - Third, the 7x overhead was calculated using a basic PyTorch setup without optimizations. Techniques like prefilling and batching could notably decrease this overhead since the instruction prompt and question are shared for 100 passages, suggesting significant potential for improved deployment efficiency.
>
> - We have also demonstrated a compelling accuracy-efficiency trade-off in Figure 5. For example, when RankRAG-8B reranks 30 contexts, the computational overhead is 2.5x that of ChatQA-1.5-8B, while its accuracy on NQ increases from 42.4 to 49.4, outperforming ChatQA-1.5-70B (NQ: 47.0) by 5%. We show the performance of RankRAG 8B with ranking 30 contexts, which consistently outperforms ChatQA 8B with an average gain of 2.3%. The gain is significant on 7 of 9 datasets.
>
> ---
> > “No significance testing was done …“
> - Research on few/zero-shot evaluation of LLMs often lacks statistical significance reporting due to two main challenges: 1) most baselines, such as PaLM 2, RA-DIT, do not release model weights, prompts and prediction results; and 2) the zero-shot performance of LLMs shows significant variance, as evidenced by the Open LLM Leaderboard, where no single model consistently leads across all datasets.
>
> - Per your suggestion, we conducted a paired statistical significance test for RankRAG against ChatQA-1.5, a strong and open-sourced baseline. Results from Fisher's randomization test are in Tables 1 (RAG) and Table 2 (Ranking) of the supplementary PDF. RankRAG significantly outperforms ChatQA-1.5 on 8/7 out of 9 datasets for the 8B/70B variants. In ranking tasks, RankRAG scores 5/4/3 out of 5 datasets for Recall@5/10/20, respectively.
>
> ---
> > “Improvements in Table 2 stem from NQ and PopQA which are relatively simple datasets that can be answered with a single context, therefore it is not apparent why RankRAG would particularly excel at those.”
>
> - While NQ and PopQA are single-hop QA datasets, some of the questions require long-tailed knowledge from Wikipedia, leading to subpar initial retrieval with a recall@5 below 75%, which is 14% lower than TriviaQA. RankRAG significantly enhances passage recall by 6%-12% by effectively reranking top passages.
>
> ---
> > "Moreover, the ranking performance for these datasets in Table 6 only marginally improves over other rerankers, therefore it is to be expected to obtain similar gains when replacing the RankRAG reranker module with other strong rerankers."
>
> - We have reported the result of RankRAG and two strong baselines using RankLlama 8B as the ranker in Table 4 of the supplementary PDF.  RankRAG outperforms these on 8 of 9 datasets, with significant gains on 6 datasets. Another advantage of RankRAG is its label and memory efficiency; it requires less labeled data for training and uses only one model during inference, unlike baselines that lack these efficiencies, while the baselines using other rerankers does not have the advantage of efficiency.
>
> ---
> > "As a side note averaging over different metrics - even though seen in many recent works - should not be done.”
>
> - Thanks for this advice. We will modify the table.
>
> ---
> > "No information about the crowdsourced dataset is provided."
>
> - In this work, we directly use the crowdsourced dataset from ChatQA work without further modification. We will include a reference to  Sec 3.2.1 of the ChatQA paper in the revision.
>
>
> ---
> > "In Section 4.2.3, the claim that LLM needs to be robust against irrelevant context contradicts the proposal to filter out irrelevant context beforehand."
>
> - Both filtering out irrelevant context and training LLMs to be robust against irrelevant context are designed to generate accurate answers. Specifically, reranking may not ensure that all top-ranked documents are relevant to the question. Enhancing the LLM's robustness to such irrelevant contexts also contributes to accurate generation. Empirically, as shown in Table 4 of the main paper, incorporating noise-robust training techniques has improved zero-shot RAG performance on these datasets by over 1%.
>
> ---
> > "It is not described how true/false tokens from context ranking are translated into a score that can be used for ranking."
> - We use the probability of the <True> token as a proxy of relevance score for ranking.
>
> ---
> > "It is unclear which number k is eventually used in RankRAG."
> - We set a fixed k=5 for RankRAG to simplify hyperparameter tuning and expedite inference. The performance of RankRAG across different k is in Figure 6, Appendix G.2.
>
> ---
> ---
> We appreciate you taking the time to review our paper again. We have added an additional comment to address the remaining questions.

---

> ### Author Response · Authors · 2024-08-07
> **Response to the remaining questions**
>
> This is a follow-up response to the remaining questions. The author's rebuttal will be released after the rebuttal deadline. It is recommended to read this response along with the rebuttal.
>
> ---
> > "The related work section could mention GRITLM"
> - Many thanks for mentioning this very relevant paper. We will cite and discuss it in our paper.
>
> ---
> > “In Section 5.1, it is not clear if baselines use different retrieval setups.”
> - Good question. we note that RankRAG and top baselines such as Llama-3-Instruct, ChatQA-1.5, InstructRetro, RA-DIT, and RePlug all employ DRAGON retriever, and some methods further finetune DRAGON (e.g. RA-DIT) for RAG applications. In main experiments of Table 2, we adopt the original setup of DRAGON as Llama3-Instruct, ChatQA-1.5 and InstructRetro to ensure a fair comparison, as RA-DIT and RePlug do not release their fine-tuned DRAGON.
> - Self-RAG uses Contriever by default. Comparing RankRAG-8b and Self-RAG-13b with the same retriever (Contriever/DRAGON) and corpus (Wikipedia), RankRAG consistently outperforms Self-RAG by 1.5%-8% on TriviaQA and PopQA, as detailed in Table 8 of the attached PDF in the author rebuttal.
>
> ---
> > “Issues in Writing: ...”
> - We appreciate the detailed comments. We will follow your advice to fix these typos.
>
> ---
> > “Why not use a dynamic number of contexts that is determined by the binary true/false label the mode already outputs?”
> - Thanks for this very interesting suggestion! We have compared this idea v.s. static $k$ on four datasets, but does not observe significant gains. Please refer to Table 7 of the attached PDF in author rebuttal for details.
>
> ---
> > “It is not sufficiently explained why one would expect a 100x increase in time.”
> - This is due to the need for an additional 100 LLM forward passes for ranking inferences. However, these inferences are much more efficient.
>
> ---
> > “Was listwise ranking also tried for context reranking?”
> - Thanks for raising this question. Indeed, we tried listwise ranking in our early experiments, but did not observe performance gains. Besides, we have also evaluated RankVicuna, a listwise ranking model in Table 6 of the supplementary PDF but found it generally performs worse than pointwise methods. Exploring how to further improve the model using listwise ranking can be an interesting future work.
>
> ---
> ---
> Thank you once again for your review.  We wish our response could address your concerns. If you have any further questions, we would be happy to discuss them further.

---

> > ### Comment · Reviewer_ze6x · 2024-08-08
> > **Additional results significantly strengthened experimental results**
> >
> > I thank the authors for addressing all of my questions and for providing extensive additional experiments that I believe strengthen the experimental results and help to back up the claims that the authors made. I am willing to update my score, given that these results find their way into the final version of the manuscript.
> >
> > The last remaining question regards the type of significance testing that was done. Why did the authors choose Fisher's randomization test over a vanilla paired t-test?

---

> ### Author Response · Authors · 2024-08-09
>
> Thank you so much for your reply and your kind words regarding the update of the score. We sincerely appreciate your constructive comments and suggestions, which greatly enhance the quality of our paper. We will incorporate all these additional results into the final version of the paper.
>
> Regarding your last question, we chose Fisher's randomization test since the distribution of test statistics in practical scenarios is often unknown and may not conform to a Gaussian distribution. Under such conditions, non-parametric tests like Fisher's randomization test are preferred for assessing statistical significance. This method is commonly employed in studies in the fields of Natural Language Processing (NLP) [1] and Information Retrieval (IR) [2].
>
> Per your question, we also tried paired t-test in the following:
>
> | Metric | NQ      | TQA       | PopQA     | HotpotQA      | 2wikimQA   | Fever    | Doc2dial | TopiOCQA | Inscit |
> |--------|---------|-----------|-----------|----------|-----------|----------|----------|----------|--------|
> | RankRAG 8B v.s. ChatQA-1.5-8B     | **7e-6** | **2e-6/1e-5** | **4e-7/1e-5** | **0.04/0.03** |**9e-7/6e-7** | **1.9e-3** | **0.03**     | 0.15     | **0.02**   |
> | RankRAG 70B v.s. ChatQA-1.5-70B    | **3e-6** | **3e-4/4e-4** | **2e-8/8e-6** | 0.12/0.08 | **1e-5/8e-6** | **4.00e-3** | 0.19     | --       | **0.03**   |
>
> From the paired t-test results, we observe that RankRAG significantly outperforms ChatQA-1.5 on the majority of datasets (8 out of 9 for 8B and 6 out of 9 for 70B).
>
> ---
> References:
>
> [1] Dror et al. "The hitchhiker’s guide to testing statistical significance in natural language processing." ACL. 2018.
>
> [2] Smuckere et al. "A comparison of statistical significance tests for information retrieval evaluation." CIKM. 2007.

---

> > ### Comment · Reviewer_ze6x · 2024-08-09
> > **No more questions**
> >
> > Thanks again for addressing my questions. I will update my score.

---

> > > ### Author Response · Authors · 2024-08-09
> > >
> > > Thank you so much! We truly enjoyed having these in-depth discussions with you.

---

### Author Rebuttal · Authors · 2024-08-07

We would like to thank all the reviewers for their thoughtful feedback.

In addition to addressing the detailed questions in each review, we have summarized the new experiments suggested by the reviewers below:

- **(ze6x, Table 1,2)** We have included the statistical significance test for both Generation (Table 1) and Ranking (Table 2) to demonstrate that the gain of RankRAG is significant.

- **(ze6x, Table 3)** We have shown additional experiments on RankRAG 8B that rank 30 contexts only, and it still consistently outperforms ChatQA-1.5 8B by a margin.

- **(ze6x, y5n5, Table 4)** We report the performance of baselines and RankRAG using RankLlama 8B as the reranker, demonstrating the advantage of using RankRAG for reranking compared to other off-the-shelf strong rerankers.

- **(y5n5, Table 5)** We have shown the performance of three OpenAI-GPT series model (GPT-3.5-turbo, GPT-4, GPT-4-turbo) with and without RAG, and demonstrated that RankRAG consistently outperforms them.

- **(ze6x, y5n5, Table 6)**  We show the performance of RankRAG over three strong ranking models (e.g., RankVicuna, RankGPT-3.5, RankGPT-4), which further justifies the efficacy of RankRAG on passage ranking tasks for RAG applications.

- **(ze6x, Table 7)** We provide an empirical comparison on the static (k=5) and dynamic number of contexts of RankRAG on four RAG tasks.

- **(ze6x, Table 8)**  We compare RankRAG and Self-RAG with the same retrieval setups and find that RankRAG consistently outperforms Self-RAG using both Contriever and DRAGON as the retrievers.


Please refer to the attached PDF for detailed information. We hope these extensive new results adequately address the concerns raised by the reviewers. If you have any further questions, please let us know; we would be happy to discuss them.

---

### Decision · Program_Chairs · 2024-09-25

**Decision:**

Accept (poster)

**Comment:**

This paper proposes RankRAG, a method to enhance Retrieval Augmented Generation by using a single LLM for both context reranking and answer generation. The authors argue that this approach improves performance, label efficiency, and memory efficiency compared to using separate models for ranking and generation. They demonstrate improved performance on 9 question-answering benchmarks, particularly in cases where the initial retrieval is suboptimal. However, the approach requires multiple times more compute than the nearest baseline, but the authors argue that the compute is worth the cost.

The reviews for the paper are positive, however I have concerns about the use of a private crowd-sourced conversational dataset. Is there a way this could not be used and achieve similar results? There are also many important experiments that are needed to contextualize the improvements of this approach over baselines added in the rebuttal (in an attached pdf) and were not in the main text. Additionally, the writing of the paper could be made more clear (as noted in the author reviews). If accepted, I hope all these points are addressed.